# Spending Thinking Time Wisely: Accelerating MCTS with Virtual Expansions

**Weirui Ye**[*]    **Pieter Abbeel**[†]    **Yang Gao** [*‡§]

[*]Tsinghua University, [†]UC Berkeley, [§] Shanghai Qi Zhi Institute

## Abstract

One of the most important AI research questions is to trade off computation versus performance since "perfect rationality" exists in theory but is impossible to achieve in practice. Recently, Monte-Carlo tree search (MCTS) has attracted considerable attention due to the significant performance improvement in various challenging domains. However, the expensive time cost during search severely restricts its scope for applications. This paper proposes the Virtual MCTS (V-MCTS), a variant of MCTS that spends more search time on harder states and less search time on simpler states adaptively. We give theoretical bounds of the proposed method and evaluate the performance and computations on $9 \times 9$ Go board games and Atari games. Experiments show that our method can achieve comparable performances to the original search algorithm while requiring less than $50\%$ search time on average. We believe that this approach is a viable alternative for tasks under limited time and resources. The code is available at `https://github.com/YeWR/V-MCTS.git`.

## 1 Introduction

When artificial intelligence was first studied in the 1950s, researchers have sought to find the solution to the question "How to build an agent with perfect rationality". The term "perfect rationality" [7, 24, 26] here refers to the decision made with infinite amounts of computations. However, one can only solve small-scale problems without considering the practical computation time since classical search algorithms usually exhibit exponential running time. Therefore, recent AI research would no longer seek to achieve "perfect rationality", but instead carefully trade-off computation versus the level of rationality. People have developed computational models like "bounded optimality" to model these settings [26]. The increasing level of rationality under the same computational budget has given us a lot of AI successes. Algorithms include the Monte-Carlo sampling algorithms, the variational inference algorithms, and using DNNs as universal function approximators [9, 8, 13, 30, 17].

Recently, MCTS-based RL algorithms have achieved much success, mainly on board games. The most notable achievement is that AlphaGo beats Hui Fan in 2015 [30]. It is the first time a computer program beat a human professional Go player. Afterward, AlphaGo beats two top-ranking human players, Lee Sedol in 2016 and Jie Ke in 2017, the latter of which ranked first worldwide at the time. Later, MCTS-based RL algorithms were further extended to other board games and Atari games [27]. EfficientZero [34] significantly improves the sample efficiency of MCTS-based RL algorithms, shedding light on its future applications in the real world like robotics and self-driving.

Despite the impressive performance of MCTS-based RL algorithms, they require massive amounts of computation to train and evaluate. For example, MuZero [27] used 1000 TPUs trained for 12 hours to learn the game of Go, and for a single Atari game, it needs 40 TPUs to train 12 hours. Compared

---

[*]`ywr20@mails.tsinghua.edu.cn, gaoyangiiis@tsinghua.edu.cn`

[†]`pabbeel@berkeley.edu`

[‡]Corresponding author

36th Conference on Neural Information Processing Systems (NeurIPS 2022).

to previous algorithms on the Atari games benchmark, it needs around two orders of magnitude more compute. This prohibitively large computational requirement has slowed down both the further development of MCTS-based RL algorithms as well as its practical use.

Under the hood, MCTS-based RL algorithms imagine the futures when taking different future action sequences. However, this imaging process for the current method is not computationally efficient. For example, AlphaGo needs to look ahead 1600 game states to place a single stone. On the contrary, top human professional players can only think through around 100-200 game states per minute [30]. Apart from the inefficiency, the current MCTS algorithm deals with easy and challenging cases with the same computational budget. However, human knows to use their time when it is most needed.

In this paper, we aim to design new algorithms that save the computational time of the MCTS-based RL methods. We make three key **contributions**: (1) We present Virtual MCTS, a variant of MCTS, to approximate the vanilla MCTS search policies with less computation. Moreover, unlike previous pruning-based methods that focus on the selection or evaluation stage in MCTS, our method improves the search loop. It terminates the search iterations earlier adaptively when current states are simpler; (2) Theoretically, we provide some error bounds of the proposed method. Furthermore, the visualization results indicate that Virtual MCTS has a better computation and performance trade-off than vanilla MCTS; (3) Empirically, our method can save more than 50% of search times on the challenging game Go $9 \times 9$ and more than 60% on the visually complex Atari games while keeping comparable performances to those of vanilla MCTS.

## 2 Related Work

### 2.1 Reinforcement Learning with MCTS

For a long time, Computer Go has been regarded as a remarkably challenging game [3, 6]. Researchers attempt to use Monte-Carlo techniques that evaluate the value of the node state through random playouts [4, 11, 12, 30]. Afterward, UCT algorithms have generally been applied in Monte-Carlo tree search (MCTS) algorithms, which use UCB1 to select action at each node of the tree [20]. Recently, MCTS-based RL methods [30, 32, 31, 27] have become increasingly popular and achieved super-human performances on board games because of their strong ability to search.

Modern MCTS-based RL algorithms include four stages in the **search loop**: selection, expansion, evaluation, and backpropagation. The computation bottlenecks in vanilla MCTS come from the search loop, especially for the evaluation stage and the selection stage of each iteration. The selection stage is time-consuming when the search tree becomes wider and deeper. The evaluation stage is quite expensive because people attempt to evaluate the node value by random playouts to the end in previous researches. Due to the search loop, MCTS-based algorithms have multiple model inferences compared to other model-free RL methods like PPO [28] and SAC [16].

### 2.2 Acceleration of MCTS

MCTS-based methods have proved their strong capability of solving complex games or tasks. However, the high computational cost of MCTS hinders its application to some real-time and more general scenarios. Therefore, numerous works are devoted to accelerating MCTS. For example, to make the selection stage more effective, some heuristic pruning methods [14, 33, 29, 1, 2] aim to reduce the width and depth of the search tree with some heuristic functions. Furthermore, for more efficient evaluations, Lorentz [22] proposed early playout termination of MCTS (MCTS-EPT) to stop the random playouts early and use an evaluation function to assess win or loss. Moreover, Hsueh *et al.* [18] applied MCTS-EPT to the Chinese dark chess and proved its effectiveness. Afterward, similar ideas have been applied in the evaluation stage of AlphaGoZero [32] and later MCTS-based methods [31, 27, 34]. They evaluate the $Q$-values through a learnable evaluation network instead of running playouts to the end. Grill *et al.* [15] propose a novel regularized policy optimization method based on AlphaZero to decrease the search budget of MCTS, which is from the optimization perspective. Danihelka *et al.* [10] propose a policy improvement algorithm based on sampling actions without replacement, named Gumbel trick to achieve better performance when planning with few simulations. However, these methods mentioned above focus on the specific stage of the search iteration or reduce the total budget through pruning and optimization methods, which are orthogonal to us. And few works targets at the search loop. Lan *et al.* [21] propose DS-MCTS, which defines the uncertainty of

MCTS and approximates it by extra DNNs with specific features for board games in training. During the evaluation, DS-MCTS will check periodically and stop the search if the state is certain.

## 3 Background

The AlphaGo series of work [30, 32, 31, 27] are all MCTS-based reinforcement learning algorithms. Those algorithms assume the environment transition dynamics are known or learn the environment dynamics. Based on the dynamics, they use the Monte-Carlo tree search (MCTS) as the policy improvement operator. I.e., taking in the current policy, MCTS returns a better policy with the search algorithm. The systematic search allows the MCTS-based RL algorithm to quickly improve the policy and perform much better in the setting where heavy reasoning is required.

### 3.1 MCTS

This part briefly introduces the MCTS method implemented in reinforcement learning applications. As mentioned in the related works, modern MCTS-based RL algorithms include four stages in the search loop, namely selection, expansion, evaluation, and backpropagation.

MCTS takes in the current states and generates a policy after the search loop of $N$ iterations. Here $N$ is a constant number of iterations set by the designer, regarded as the total budget. In the selection stage of each iteration, an action will be selected by maximizing over UCB. Specifically, AlphaZero [31] and MuZero [27] are developed based on a variant of UCB, P-UCT [25] and have achieved great success on board games and Atari games. The formula of P-UCT is the Eq (1):

$$a^k = \arg\max_{a \in \mathcal{A}} Q(s,a) + P(s,a) \frac{\sqrt{\sum_{b \in \mathcal{A}} N(s,b)}}{1 + N(s,a)} (c_1 + \log((\sum_{b \in \mathcal{A}} N(s,b) + c_2 + 1)/c_2)), \quad (1)$$

where $k$ is the index of iteration, $\mathcal{A}$ is the action set, $Q(s,a)$ is the estimated Q-value, $P(s,a)$ is the policy prior obtained from neural networks, $N(s,a)$ is the visitations to select the action $a$ from the state $s$ and $c_1, c_2$ are hyper-parameters. The output of MCTS is the visitation of each action of the root node. After $N$ search iterations, the final policy $\pi(s)$ is defined as the normalized root visitation distribution $\pi_N(s)$, where $\pi_k(s,a) = N(s,a)/\sum_{b \in \mathcal{A}} N(s,b) = N(s,a)/k, a \in \mathcal{A}$. For simplification, we use $\pi_k$ in place of $\pi_k(s)$ sometimes. And the detailed procedure of MCTS is introduced in Appendix. In our method, we propose to approximate the final policy $\pi_N(s)$ with $\hat{\pi}_k(s)$, which we name as a virtual expanded policy, through a new expansion method and a termination rule. In this way, the number of iterations in MCTS can be reduced from $N$ to $k$.

### 3.2 Computation Requirement

Most of the computations in MCTS-based RL are in the MCTS procedure. Each action taken by MCTS requires $N$ times neural network evaluations, where $N$ is a constant number of iterations in the search loop. Traditional RL algorithms, such as PPO [28] or DQN [23], only need a single neural network evaluation per action. Thus, MCTS-based RL is roughly $N$ times computationally more expensive than traditional RL algorithms. In practice, training a single Atari game needs 12 hours of computation time on 40 TPUs [27]. The computation need is roughly two orders of magnitude more than traditional RL algorithms [28], although the final performance of MuZero is much better.

## 4 Method

We aim to spend more search time on harder states and less on easier states. Intuitively, human knows when to make a quick decision or a slow decision under different circumstances. Unfortunately, this situation-aware behavior is absent in current MCTS algorithms. Therefore, we propose an MCTS variant that terminates the search iteration adaptively. It consists of two components: a novel expansion method named virtual expansion to estimate the final visitation based on the current partial tree; a termination rule that decides when to terminate based on the hardness of the current scenario. And we will display the adaptive mechanism through visualizations in Section 5.5.

## 4.1 Termination Rule

We propose to terminate the search loop earlier based on the current tree statistics. Intuitively, we no longer need to search further if we find that recent searches have little changes on the root visitation distribution. With this intuition in mind, we propose a simple modification to the MCTS search algorithm. As mentioned in 3.1, $\pi_k(s)$ is the policy defined by the visitations of the root state at iteration $k$. Let $\Delta_s(i, j)$ be the L1 difference of $\pi_i(s), \pi_j(s)$, namely $\Delta_s(i, j) = ||\pi_i(s) - \pi_j(s)||_1$. Then we terminate the search loop when we have searched at least $rN$ iterations and $\Delta_s(k, k/2) < \epsilon$, where $\epsilon$ is a tolerance hyper-parameter, $r \in (0, 1)$ is the ratio of the minimum search budget and $N$ is the full search iterations. We show that under certain conditions, a bound on $\Delta_s(k, k/2)$ implies a bound on $\Delta_s(k, N)$. $\Delta_s(k, N)$ measures the distance between the current policy $\pi_k(s)$ and the oracle policy $\pi_N(s)$. In this way, $\Delta_s(k, k/2)$ reflects the hardness of the state $s$. Consequently, once the gap is small enough, it is unnecessary for more search iterations.

## 4.2 Virtual Expansion in MCTS

| **Algorithm 1** Iteration of vanilla MCTS | **Algorithm 2** Iteration of MCTS with **Virtual Expansion** |
|---|---|
| 1: Current $k$-th iteration step: | 1: Current $k$-th iteration step: |
| 2: **Input:** $\mathcal{A}, P, Q_k(s,a), N_k(s,a)$ | 2: **Input:** $\mathcal{A}, P, Q_k(s,a), N_k(s,a), \hat{N}_k(s,a)$ |
| 3: Initialize: $s \leftarrow s_{\text{root}}$ | 3: |
| 4: **repeat** do search | 4: **if** Not init $\hat{N}_k(s,a)$ **then** |
| 5:     $a^* \leftarrow \text{UCB1}(Q, P, N)$ | 5:     Init: $\hat{N}_k(s,a) \leftarrow N_k(s,a)$ |
| 6:     $s \leftarrow$ next state$(s, a^*)$ | 6: **end if** |
| 7: **until** $N_k(s, a^*) = 0$ | 7: |
| 8: Evaluate the state value $R(s,a)$ and $P(s,a)$ | 8: $s \leftarrow s_{\text{root}}$ |
| 9: **for** $s$ along the search path **do** | 9: $a^* \leftarrow \text{UCB1}(Q, P, \hat{N})$ |
| 10:     $Q_{k+1}(s,a) = \frac{N_k(s,a) \cdot Q_k(s,a) + R(s,a)}{N_k(s,a)+1}$ | 10: $\hat{N}_k(s,a) \leftarrow \hat{N}_k(s,a) + 1$ |
| 11:     $N_{k+1}(s,a) = N_k(s,a) + 1$ | 11: |
| 12: **end for** | 12: **Return** $\hat{N}_k(s,a)$ |
| 13: **Return** $Q_{k+1}(s,a), N_{k+1}(s,a)$ | |

For the termination rule $\Delta_s(k, k/2) < \epsilon$, we assume $\pi_i$ and $\pi_j$ are directly comparable. However, they are not directly comparable because the tree is expanded with UCT. As the number of visits increases, the upper bound would be tighter, and the latter visits are more focused on the promising parts. Thus earlier visitation distributions (smaller iteration number) can exhibit more exploratory distribution, while latter ones (larger iteration number) are more exploitative on promising parts.

To compare $\pi_i$ and $\pi_j$ properly, we propose a method called **virtual expansion** in place of the vanilla expansion. Briefly, it aligns two distributions by virtual UCT expansions until the constant budget $N$. When the tree is expanded at iteration $k$, it has $N - k$ iterations to go. A normal expansion would require evaluating neural network $N - k$ times for a more accurate $Q(s, a)$ estimate for each arm at the root node. Our proposed virtual expansion still expands $N - k$ times according to UCT, but it ignores the $N - k$ neural network evaluations and assumes that each arm's $Q(s, a)$ does not change. We denote the virtual expanded distribution from $\pi_i$ as a **virtual expanded policy** $\hat{\pi}_i$. By doing virtual expansions on both $\pi_i$ and $\pi_j$, we will obtain the corresponding virtual expanded policies $\hat{\pi}_i, \hat{\pi}_j$. Here we effectively remove the different levels of exploration/exploitation in the two policies. Then the termination condition becomes the difference of virtual expanded policies. We name the rule as **VET-Rule** (Virtual Expanded Termination Rule):

$$\hat{\Delta}_s(k, k/2) = \left|\left|\hat{\pi}_k(s) - \hat{\pi}_{k/2}(s)\right|\right| < \epsilon. \tag{2}$$

The comparisons between vanilla expansion and virtual expansion are illustrated in Algorithm 1 and 2. The time-consuming computations are highlighted in Algorithm 1. Line 4 to 7 in Algorithm 1 target at searching with UCT to reach an unvisited state for exploration. Then it evaluates the state and backpropagates along the search path to better estimate $Q$-values. After total $N$ iterations, the visitation distribution of the root node $\pi_N(s)$ is considered as the final policy $\pi(s)$. However, in virtual expansion, listed in Algorithm 2, it only searches one step from the root node. And it selects actions based on the current estimations without changing any properties of the search

tree. Furthermore, the virtual visited counts $\hat{N}_k(s,a)$ are changed after virtual visits to balance the exploitation and the exploration issue. After $N - k$ times virtual expansion, the virtual expanded policy becomes $\hat{\pi}_k(s,a) = \hat{N}_k(s,a)/N$ instead of $N_k(s,a)/k$. When $k = N$, further searches after the root have no effects on the final policy. So $\hat{\pi}_N(s,a) = \pi_N(s,a)$.

## 4.3  V-MCTS Algorithm

The procedure of MCTS with VET-Rule is listed in Algorithm 3. We name our method **Virtual MCTS** (V-MCTS), a variant of MCTS with VET-Rule. Compared with the original MCTS, lines 8-13 are the pseudo-code for the rule. In each iteration, we do some calculations with little cost to judge whether the VET-Rule is satisfied. If it is, then the search process is terminated and returns the current virtual expanded policy $\hat{\pi}_k(s)$. Thus, it skips the next $N - k$ model predictions from neural networks in the evaluation stage highlighted in line 7. In this way, we can approximate the oracle distribution $\pi_N$ by $\hat{\pi}_k$ while reducing the budget of $N$ simulations to $k$. Here, $k \geq rN$ and $r$ is a hyperparameter of the minimum budget $rN$. We can reduce the tree size by $1/r$ times at most.

---

**Algorithm 3** Virtual MCTS

---

1: Input: budget $N$, state $s$, conservativeness $r$, error $\epsilon$
2: Notice: $\pi_k(s), \hat{\pi}_k(s)$ are policy distributions.
3: Notice: $\pi_k(s,a), \hat{\pi}_k(s,a)$ are probabilities for action $a$.
4: **for** $k \in N$ **do**
5:     Selection with UCB1
6:     Expansion for the new node
7:     Evaluation  with Neural Networks
8:     Backpropagation for updating Q and visitations
9:     $\pi_k(s,a) \leftarrow N_k(s,a)/k$
10:    Virtual expand $N - k$ nodes and update $\hat{N}(s,a)$
11:    $\hat{\pi}_k(s,a) \leftarrow \hat{N}_k(s,a)/N$
12:    **if** $k \geq rN \wedge \left\| \hat{\pi}_k(s) - \hat{\pi}_{k/2}(s) \right\|_1 < \epsilon$ **then**
13:       $\pi(s) \leftarrow \hat{\pi}_k(s)$
14:       **Break**
15:    **end if**
16:    $\pi(s) \leftarrow \pi_k(s)$
17: **end for**
18: **Return** $\pi(s)$

---

## 4.4  Theoretical Justifications

Furthermore, we will give some theoretical bounds on the $Q$-values and $\hat{\Delta}_s(k,N)$ of V-MCTS. Before this, we define some notations first: $k$ is the index of the current search iteration, and $N$ is the number of total search iterations. $\mathcal{A}$ is the action set and $|\mathcal{A}| > 1$. Each action $a \in \mathcal{A}$ is associated with a value, which is a random variable bounded in the interval $[0, 1]$ with expectation $Q_a$. At step $k$, the empirical mean value over the $k$ trails is $Q_k(s,a)$. For simplification, we denote $Q_k(s,a)$ as $\bar{Q}_a^k$. We denote the empirical mean value after $N - k$ virtual expansion as $\hat{Q}_a^N$. Since we only deal with the visitation distribution of the root, we omit the state subscript for the root state. For convenience, we assume that different actions are ordered by their corresponding expected values, which means that $1 \geq Q_1 \geq Q_2 \geq \cdots \geq Q_a \geq \cdots \geq Q_{|\mathcal{A}|} \geq 0$.

**Theorem 4.1.** *Given* $r \in (0,1)$, *confidence* $\delta \in (0,1)$, *finite action set* $\mathcal{A}$. $\exists N_0 > 0, \forall N > N_0, k \geq rN$, *let* $\epsilon_k = \sqrt{\frac{1}{2k} \ln \frac{100k^2}{\delta}}$, *after* $k$ *times vanilla expansion and* $N - k$ *times virtual expansion, we have (a)* **Value Consistency in Virtual Expansion:** $Pr\{\bigcap_{a \in \mathcal{A}} \left| \hat{Q}_a^N - Q_a \right| < \epsilon_k\} > (1 - \frac{e\delta|\mathcal{A}|}{50r^2 N^2})$; *(b)* **Best Action Identification in Virtual Expansion:** $Pr\{\left| \hat{Q}_*^N - \bar{Q}_1^N \right| < \epsilon_k + \epsilon_N\} > 1 - 2\left(\frac{\delta}{50k^2} \exp\left(\frac{1}{1.61\sqrt{k}}\right) + \frac{\delta}{50N^2} \exp\left(\frac{1}{N}\right)\right)$, *where* $* := \arg\max_{a \in \mathcal{A}} \bar{Q}_a^k$, $e$ *is the Euler's number.*

Theorem 4.1 (a) gives a bound of the distance between the empirical mean values after virtual expansions and the expected values. Noticed that $\lim_{N \to \infty} \epsilon_{rN} = 0$ and $\lim_{N \to \infty} \frac{e\delta|\mathcal{A}|}{50r^2 N^2} = 0$. It tells that, after enough trails, the expected $Q$-values of all actions can be estimated by the corresponding empirical $Q$-values after virtual expansion. Furthermore, when the $Q$-values have converged, the effect of virtual expansion is the same as that of vanilla expansion. Denote the best empirical action as $*$, and the best expected action is 1 because $Q_1 \geq Q_a$. Theorem 4.1 (b) notes that the $Q$-value of the best empirical action with virtual expansion is of high probability to be close to the Q-value of the best expected action with vanilla expansion. Intuitively, it tells that whether or not we successfully find the best expected action, the best empirical action has similar effects to the best

expected action. And, $N_0$ should be larger than the action space size, otherwise it cannot satisfy the theorem conditions. The proof is attached in Appendix.

**Theorem 4.2.** *(**Error Bound of V-MCTS**): Given $r \in (0, 1)$, confidence $\delta \in (0, 1)$, finite action set $\mathcal{A}$. Suppose the virtual expanded policy $\hat{\pi}_k$ is generated from Algorithm 3 (V-MCTS), $\exists N_0 > 0$, $\forall N > N_0, k \geq rN$, $\forall \epsilon \in (0, 1]$, if $\left\| \hat{\pi}_k(s) - \hat{\pi}_{k/2}(s) \right\|_1 < \epsilon$, we have $Pr\{ \|\pi_N(s) - \hat{\pi}_k(s)\|_1 < 3\epsilon \} > 1 - \frac{e\delta|\mathcal{A}|}{50N^2}(1 + \frac{4}{r^2})$, where $e$ is the Euler's number.*

Theorem 4.2 tells that a bound of $\hat{\Delta}_s(k, k/2)$ implies a bound of $\hat{\Delta}_s(k, N)$ with high probability. Noticed that $\lim_{N \to \infty} \frac{e\delta|\mathcal{A}|}{50N^2}(1 + \frac{4}{r^2}) = 0$. Therefore, the oracle policy $\pi_N(s)$ can be approximated by $\hat{\pi}_k(s)$ after enough trails. The proof of this theorem is attached in Appendix.

Given the minimum distance $\epsilon$, for easier states, the rule $\hat{\Delta}_s(k, k/2) < \epsilon$ is easier to satisfy. That's because the Q-values of the tree nodes keep in a small range even with more search iterations. Thus, the virtual expanded policy generated by V-MCTS is close to the oracle policy, and the search loop will be terminated earlier if the state is easier. In the next section, we do ablations to investigate the effects of the hyper-parameters and show visualizations to verify the adaptive behavior.

## 5  Experiments

In this section, the goal of the experiments is to prove the effectiveness and efficiency of V-MCTS. First, we compare the performance and the cost between the vanilla MCTS and our method. Specifically, we evaluate the board game Go $9 \times 9$, and a few Atari games. In addition, we do some ablations to examine the virtual expansion's effectiveness and evaluate the sensitiveness of hyper-parameters. Finally, we show the adaptive mechanism through visualizations and performance analysis.

### 5.1  Setup

**Models and Environments** Recently, Ye *et al.* [34] proposed EfficientZero, a variant of MuZero [27] with three extra components to improve the sample efficiency, which only requires 8 GPUs in training, and thus it is more affordable. Here we choose the board game Go $9 \times 9$ and a few Atari games as our benchmark environments. The game of Go tests how the algorithm performs in a challenging planning problem. And Atari games feature visual complexity.

**Hyper-parameters** As for the Go $9 \times 9$, we choose Tromp-Taylor rules. The environment of Go is built based on an open-source codebase, GymGo [19]. We evaluate the performance of the agent against GNU Go v3.8 at level 10 [5] for 200 games. We include 100 games as the black player and 100 games as the white one with different seeds. We set the komi to 6.5, as most papers do. As for the Atari games, we choose 5 games with 100k environment steps. In each setting, we use 3 training seeds and 100 evaluation seeds for each trained model. More details are attached in Appendix.

**Baselines** We compare our method to EfficientZero with vanilla MCTS, on Go $9 \times 9$ and some Atari games. Moreover, DS-MCTS [21] also terminates the MCTS adaptively through trained uncertainty networks. But it requires specific features designed for Go games, and it only works in the evaluation stage. Therefore, we also compare the final performance for Go games with the DS-MCTS.

### 5.2  Results on Go

Figure 1(a) illustrates the computation and performance trade-off on Go against the same GnuGo (level 10) agent. The x-axis is the training speed, and the y-axis is the winning rate. Therefore, the curve which lies to the top-left has better performance than the bottom-right in terms of the trade-off. We train the baseline method with constant budgets $N$, which is noted as the blue points. Besides, we also train the V-MCTS with hyperparameters $r = 0.2, \epsilon = 0.1$. We evaluate the trained model with different $\epsilon$ to display the trade-off between computation and performance, indicated as the red points. And the green points are the GnuGo with different levels. The GnuGo engine provides models of different levels (1-10). Each level is a trade-off between the run time and the strength of the agent. The y-axis is the winning rate against the model of level 10. Here the green curve shows the performance-computation trade-off of the GnuGo engine.

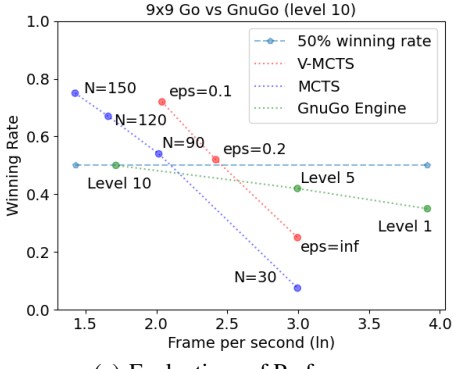
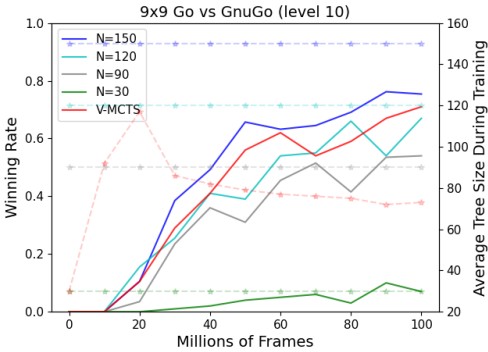

(a) Evaluations of Performance        (b) Wining Rates and Tree Size during Training

Figure 1: Performance of Virtual MCTS on Go $9 \times 9$ against GnuGo (level 10). (a) Evaluating the speed and winning rate of MCTS, V-MCTS, and GnuGo at different levels. V-MCTS has better computation and performance trade-off. X-axis is the frame per second in the ln scale. (For convenience, the eps in Figure (a) denotes the hyperparameter $\epsilon$.) (b) Evaluating the winning rate and the average tree size during the training stage. The **solid lines** and **dashed lines** display the winning rate and the tree size, respectively. The red one is V-MCTS, and the others are vanilla MCTS with different $N$. V-MCTS makes the tree size adaptive in training and reduces the search cost while performing comparably to the vanilla MCTS ($N = 150$). However, reducing $N$ in vanilla MCTS results in a more significant performance drop.

Firstly, for all the methods, more search iterations (larger $N$ or smaller $eps$) lead to higher winning rates but result in more response time. Secondly, V-MCTS ($\epsilon = 0.1$) achieves **71%** winning rate against the GnuGo level 10, which is close to 75% from MCTS ($N = 150$). And the time cost of V-MCTS ($\epsilon = 0.1$) for a one-step move is **0.12s** while the GnuGo engine is 0.18s and MCTS ($N = 150$) is more than 0.2s. Therefore, such termination rule can keep strong performances with less budget. For a more detailed breakdown of the time consumption of V-MCTS, please see the Appendix. Finally, we can find that the red dashed line lies to the right of the blue one. It indicates that V-MCTS is better than the vanilla MCTS considering the computation and performance trade-off.

Figure 1(b) illustrates the changes of winning rates and the average tree size over the training stage. Here, as the red dashed line shows, the tree size of V-MCTS varies over training and keeps smaller than the maximum size ($N = 150$) while the winning rate keeps comparable to the MCTS ($N = 150$). Consequently, V-MCTS can work well not only in evaluation but also in training.

We also compare our method to DS-MCTS [21], which terminates the search when the state is predicted to be certain with DNNs. To make fair comparisons, we implement the DS-MCTS and follow their design of features for Go

Table 1: Results for Go $9 \times 9$: Comparison of the winning rate and the average budget over 200 games for 3 separate training runs.

|  | MCTS (N=150) | DS-MCTS | V-MCTS |
|---|---|---|---|
| Average budget | $150 \pm 0.0$ | $97 \pm 12.5$ | $\mathbf{76} \pm 10.8$ |
| Winning rate | $\mathbf{75\%} \pm 3.0\%$ | $60\% \pm 4.0\%$ | $71\% \pm 4.7\%$ |

games. We set $N_{max} = 150, c = \{30, 75, 120\}, thr = \{.1, .1, .1\}$ in DS-MCTS. Then we compare the winning rate and the average budget among the vanilla MCTS, DS-MCTS, and V-MCTS.

Table 2: Results for Atari games: scores over 100 evaluation seeds for 3 separate training runs. $k$ is the average budget of V-MCTS. MCTS ($N = 50$) is the oracle one. The best results among distinct versions except the oracle are in bold. V-MCTS achieves better performance-computation trade-off.

| MCTS | $N = 50$ | $N = 30$ | $N = 10$ | **Ours (V-MCTS)** | **Budget $k$** |
|---|---|---|---|---|---|
| Pong | $19.7 \pm 1.6$ | $12.5 \pm 5.5$ | $2.0 \pm 1.3$ | $\mathbf{18.8} \pm 2.8$ | $13.3 \pm 0.6$ |
| Breakout | $410.7 \pm 15.1$ | $370.9 \pm 34.1$ | $303.9 \pm 11.3$ | $\mathbf{372.8} \pm 18.3$ | $15.7 \pm 0.6$ |
| Seaquest | $1159.9 \pm 90.7$ | $775.2 \pm 146.8$ | $555.4 \pm 66.9$ | $\mathbf{970.0} \pm 339.5$ | $14.3 \pm 1.2$ |
| Hero | $9992.1 \pm 2059.4$ | $\mathbf{9241.3} \pm 3615.3$ | $4437.0 \pm 2490.6$ | $8928.1 \pm 2922.1$ | $15.0 \pm 1.0$ |
| Qbert | $14495.8 \pm 683.9$ | $10429.9 \pm 2291.1$ | $8149.8 \pm 2085.0$ | $\mathbf{11476.6} \pm 978.2$ | $16.3 \pm 1.2$ |

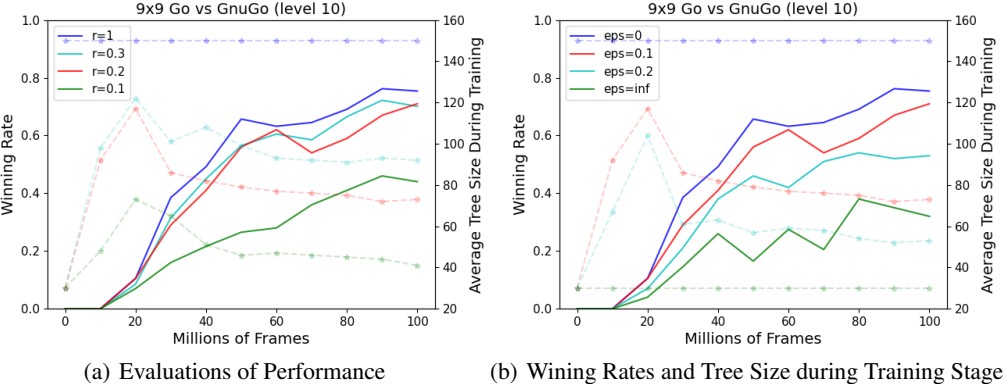

(a) Evaluations of Performance          (b) Wining Rates and Tree Size during Training Stage

Figure 2: Sensitivity of VET-Rule to the hyperparameter $r, \epsilon$ on Go $9 \times 9$. The solid lines and dashed lines display the winning probability and the average tree size, respectively.

Experiments show that V-MCTS outperforms the DS-MCTS in both aspects, listed in Table 1. We attribute the better performance of V-MCTS to the virtual expanded policy. DS-MCTS chooses $\pi_k(s)$ as the policy after the termination of the search, while V-MCTS chooses $\hat{\pi}_k(s)$, which has theoretical guarantees to approximate $\pi_N(s)$.

## 5.3 Results on Atari

Apart from the results of Go, we also evaluate our method on some visually complex games. Since the search space of Atari games is much smaller than that of Go and the Atari games are easier, we choose a few Atari games to study how the proposed method impacts the performance. We follow the setting of EffcientZero, 100k Atari benchmark, which contains only 400k frames data. The results are shown in Table 2. Generally, we find that our method works on Atari games. The tree size is adaptive, and the performance of V-MCTS is still comparable to the MCTS with full search trails. It has better performance than the MCTS($N = 30$) while requiring fewer searches, proving the effectiveness and efficiency of our proposed method. The Hero game is an outlier here. But our performance is very close to the vanilla MCTS ($N = 30$) while we use half of the search iterations on average. Besides, the number of search times decreases more than that on Go.

To sum up, V-MCTS can keep comparable performance under fewer search iterations while simply reducing the total budget of MCTS will encounter a more significant performance drop. In addition, the savings of search cost is more substantial in easier environments.

## 5.4 Ablation Study

The results in the previous section suggest that our method reduces the response time of MCTS while keeping comparable performance on challenging tasks. This section tries to figure out which component contributes to the performance and how the hyperparameters affect it. And we also ablate the effects of different normalization criterions in VET-Rule and the larger budget ($N$) in MCTS.

**Virtual Expansion** In Section 4.2, we introduce the virtual expansion. To prove the effectiveness of virtual expansion, we compare it with another two baseline expansion methods. One is the vanilla expansion, mentioned in Algorithm 1, which returns at iteration $k$ and outputs $\pi_k$. Another is greedy expansion, which spends the left $N - k$ simulations in searching the current best action greedily, indicating that $\hat{\pi}_k(s, a) = (N_k(s, a) + (N - k)\mathbf{1}_{b=\arg\max N_k(s,b)})/N$. Briefly, we stop the search process after $k = 30$ iterations and do $N - k$ times virtual expansion or greedy expansion or nothing, where $k = rN$ and $r = 0.2, N = 150$.

We compare the winning rate against the same engine, and the results are listed as Table 3 shows. The winning rate of virtual expansion can achieve 32%, which is much better than the others. Besides, MCTS with greedy expansion does not work because it over-exploits and results in severe exploration issues. Consequently, virtual expansion can generate a better policy distribution because it can balance exploration and exploitation with UCT.

**Termination Rule** It is significant to explore a better termination rule to keep the sound performance while decreasing the tree size as much as possible. As mentioned in Section 4.1, VET-Rule has two hyperparameters $r, \epsilon$. Here $r$ is the factor of the minimum budget $rN$, and $\epsilon$ is the minimum distance $\hat{\Delta}_s(k, k/2)$. To explore the VET-Rule with better computation and performance trade-off, we do ablations for the different values of $r$ and $\epsilon$, respectively. The default values of $r, \epsilon$ are set to $0.2, 0.1$.

Table 3: Ablation results of different expansion methods on Go $9 \times 9$ for 3 separate training runs.

| Algorithm | Size Avg. | Winning Rate |
|---|---|---|
| Vanilla expansion | 30 | $17\% \pm 3.2\%$ |
| Greedy expansion | 30 | $3\% \pm 2.0\%$ |
| Virtual expansion | 30 | $\mathbf{32\%} \pm 3.5\%$ |

Figure 2 compares the winning rate as well as the average tree size across the training stage. Firstly, Figure 3(a) gives the results of different minimum search times factor $r$. The winning probability is not sensitive to $r$ when $r \geq 0.2$. Nevertheless, the average tree size is sensitive to $r$ because V-MCTS is supposed to search for at least $rN$ times. In addition, there is a performance drop between $r = 0.1$ and $r = 0.2$. Therefore, it is reasonable to choose $r = 0.2$ to balance the speed and the performance.

Besides, the comparisons of the different minimum distance $\epsilon$ are shown in Figure 3(b). A larger $\epsilon$ makes the tree size smaller because $\hat{\Delta}_s(k, k/2) < \epsilon$ is easier to satisfy. In practice, the performance is highly correlated with $\epsilon$. In terms of the winning rate, a smaller $\epsilon$ outperforms a larger one. However, better performances are at the cost of more computations. We suggest selecting an appropriate minimum distance to balance the computation and performance ($r = 0.2, \epsilon = 0.1$).

**Normalization criterion in VET-Rule** The proposed VET-Rule, $\left\|\hat{\pi}_k(s) - \hat{\pi}_{k/2}(s)\right\| < \epsilon$ is a termination condition for V-MCTS. And L2 norm is another reasonable choice to amplify the bigger deviations. Therefore, we make ablations of the normalization criterion for the policy distributions. Specifically, we take a pretrained model, and compare the different strategies of L1 norm and L2 norm, namely, $\left\|\hat{\pi}_k(s) - \hat{\pi}_{k/2}(s)\right\|_1 < \epsilon$ and $\left\|\hat{\pi}_k(s) - \hat{\pi}_{k/2}(s)\right\|_2 < \epsilon$. The results are as Tab. 4 shows. We can find that (1) L2 norm can also work for V-MCTS; (2) L1 norm is better than L2 norm. And we attribute this to the formulation of ucb scores. Because the ucb scores have already taken into account the difference in the visitations (see the N(s, a) in Eq (1)). Therefore, amplifying the deviations may result in some bias.

Table 4: Comparison of the winning rate and the average budget with different norm strategies in VET-Rule. L1 Norm means $\left\|\hat{\pi}_k(s) - \hat{\pi}_{k/2}(s)\right\|_1 < \epsilon$ and L2 Norm means $\left\|\hat{\pi}_k(s) - \hat{\pi}_{k/2}(s)\right\|_2 < \epsilon$.

| | Average budget | Winning rate |
|---|---|---|
| MCTS ($N = 150$) | 150 | 82.0% |
| V-MCTS **L1** Norm, $N = 150, r = 0.2, \epsilon = 0.1$ | **96.2** | **81.5%** |
| V-MCTS **L2** Norm, $N = 150, r = 0.2, \epsilon = 0.1$ | 97.1 | 79.8% |
| V-MCTS **L2** Norm, $N = 150, r = 0.2, \epsilon = 0.05$ | 119.3 | 81.0% |

**Larger budget ($N$) in MCTS** To investigate whether our method still holds with larger amounts of MCTS expansions, we take a pretrained model and compare two strategies: (1) vanilla expansion with N=150/400/600/800 nodes in MCTS (2) virtual expanded policy with $N = 800, r = 0.2, \epsilon = 0.1$. The results are listed in Tab. 5. The result shows that (1) V-MCTS($N = 800, r = 0.2, \epsilon = 0.1$) is better than MCTS ($N = 600$) in both the average budget and the winning rate, (2) V-MCTS can achieve comparable performance to the oracle MCTS($N = 800$) while keeping much less average budget. Therefore, V-MCTS works with a larger amount of MCTS expansions.

Table 5: Comparison of the winning rate and the average budget with larger amounts of MCTS expansions. Here the hyper-parameters of our method are $N = 800, r = 0.2, \epsilon = 0.1$.

| **MCTS** | $N = 150$ | $N = 400$ | $N = 600$ | $N = 800$ | Ours |
|---|---|---|---|---|---|
| Average budget | 150 | 400 | 600 | 800 | 431.1 |
| Winning rate | 82.0% | 84.5% | 84.9% | 85.9% | 85.0% |

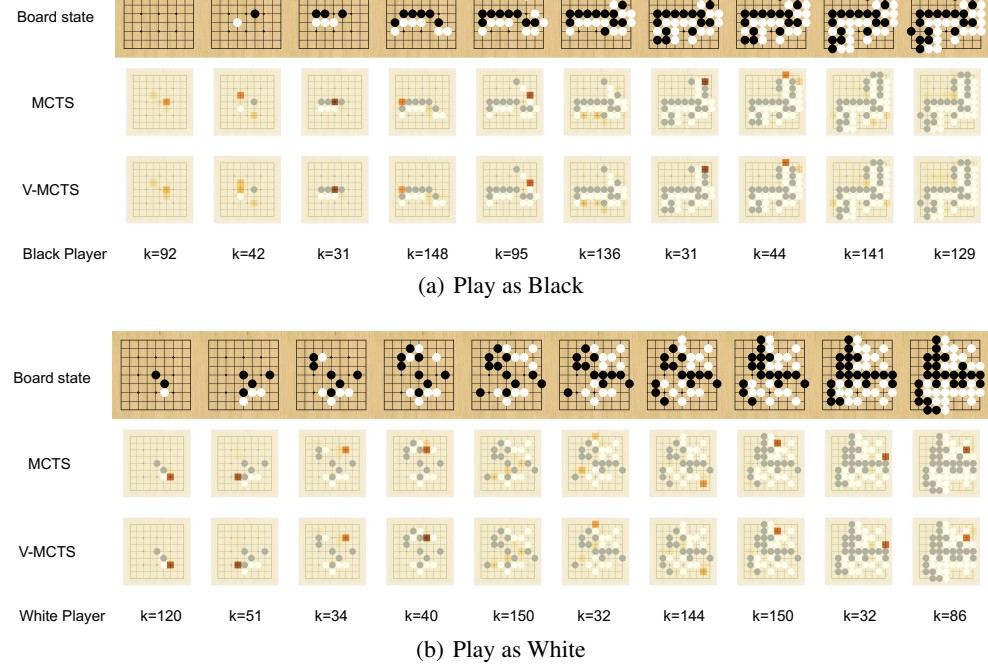

Figure 3: Heatmap of policy distributions from the MCTS ($N = 150$) and the V-MCTS. The agent play as Black in (a) and White in (b) against the GnuGo (level 10). Our agent wins in both of the games. A darker red color represents larger visitations of the corresponding action. The V-MCTS will terminate with different search times $k$ according to the situations and generate a near-oracle policy distribution.

## 5.5 Visualization of V-MCTS's Adaptive Behavior

In this section, we display the adaptive mechanism through visualizations and performance analysis. We find that (1) $\hat{\pi}_k(s)$ is close to $\pi_N(s)$; (2) V-MCTS terminates earlier for simpler states.

Specifically, we choose some states at different time steps on one game of Go against the GnuGo with a trained model. And Figure 3 is the visualization of the policy distributions heatmap. Here we add two games, which contain one black player and one white player. The last two rows in each subfigure are the heatmap visualization for oracle MCTS ($\pi_N$) and V-MCTS ($\hat{\pi}_k$ when $\hat{\Delta}_s(k, k/2) < \epsilon$). The darker the color is on the grid, the more the corresponding action is visited during the search. In general, $\hat{\pi}_k$ is close to the $\pi_N$ at distinct states, indicating that the virtual expanded policy obtained after virtual expansion is close to the oracle one.

Furthermore, the less valuable actions there are, the sooner the V-MCTS will terminate. For example, on Go games, the start states are usually not complex because there are only a few stones on the board, but the situations are more complicated in shuban, the closing stage of the game. Notably, the termination occurs earlier in the start states (columns 1, 2, 3), but it is the opposite when the situation is more complicated. More importantly, the termination step $k$ is not related to the number of Go pieces. Therefore, we can conclude that V-MCTS makes adaptive terminations according to the situations of the current states and generate near-oracle policies. Specifically, it terminates the search loop earlier when handling easier states, which has a better computation and performance trade-off.

## 6 Discussion

This paper proposes a novel method named V-MCTS to accelerate the MCTS to determine the termination of search iterations. It can maintain comparable performances while reducing half of the time to search adaptively. We believe that this work can be one step toward applying the MCTS-based methods to some real-time domains. One limitation of our work is that it cannot deal with the environments of continuous action space. In the future, we will plan to extend to the continuous action space with early termination.

## Acknowledgments and Disclosure of Funding

This work is supported by the Ministry of Science and Technology of the People´s Republic of China, the 2030 Innovation Megaprojects "Program on New Generation Artificial Intelligence" (Grant No. 2021AAA0150000). This work is also supported by a grant from the Guoqiang Institute, Tsinghua University.

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



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
