# A  Appendix

## A.1  Experimental setup

### A.1.1  Models and Hyper-parameters

**MCTS in modern RL algorithms** As mentioned in the related works, modern MCTS-based RL algorithms include four stages in the search loop, namely selection, expansion, evaluation, and backpropagation. (1) The selection stage targets selecting a new leaf node with UCT. (2) The expansion stage expands the selected node and updates the search tree. (3) The evaluation stage evaluates the value of the new node. (4) The backpropagation stage propagates the newly computed value to the nodes along the search path to obtain more accurate Q-values with Bellman backup.

**Model Design** As for the architecture of the networks, we follow the implementation of EfficientZero [34] in Atari games, which proposes three components based on MuZero: self-supervised consistency, value prefix, and off-policy correction. In the implementation of EfficientZero, there is a representation network, a dynamics network, and a reward/value/policy prediction network. The representation network is to encode observations to hidden states. The dynamics network is to predict the next hidden state given the current hidden state and an action. The reward/value/policy prediction network is to predict the reward/value/policy. Notably, they propose to keep temporal consistency between $s_{t+1}$ and the predicted state $\hat{s}_{t+1}$. The training objective is:

$$\mathcal{L}_t(\theta) = \lambda_1 \mathcal{L}(u_t, r_t) + \lambda_2 \mathcal{L}(\pi_t, p_t) + \lambda_3 \mathcal{L}(z_t, v_t) + \lambda_4 \mathcal{L}_{\text{similarity}}(s_{t+1}, \hat{s}_{t+1}) + c||\theta||^2$$

$$\mathcal{L}(\theta) = \frac{1}{l_{\text{unroll}}} \sum_{i=0}^{l_{\text{unroll}}-1} \mathcal{L}_{t+i}(\theta), \tag{3}$$

where $u_t, \pi_t, z_t$ are the target reward/policy/value of the state $s_t$ and $r_t, p_t, v_t$ are the predicted reward/policy/value of the state $s_t$ respectively. The prediction will do $l_{\text{unroll}} = 5$ times iteratively for the state $s$ on both Go and Atari games. We do some changes when dealing with board games. Significantly, we remove the reward prediction network because the agent will receive a reward only at the end of the games. The other major changes for board games are listed as follows.

Since the board game Go is harder than the Atari games, we add more residual blocks (two times blocks). Specifically, we use 2 residual blocks in the representation network, the dynamics network, as well as the value/policy prediction network on Go $9 \times 9$ while EfficientZero uses only 1 residual block in those networks on Atari games. As for the representation network, we remove the downsampling part here because there is no need to do downsampling for Go states. In the value/policy prediction networks, we enlarge the dimension of the hidden layer from 32 to 128. Besides, considering that the reward is sparse on Go (only the final value) and the collected data are sufficient, we only take the self-supervised consistency component in EfficientZero to give more temporal supervision during training.

**Hyper-parameters** In each case, we train EfficientZero for unrolled 5 steps and mini-batches of size 256. Besides, the model is trained for 100k batches with 100M frames of data on board games while 100k batches with 400k frames in Atari games. We stack 8 frames in board games without frameskip while stacking 4 frames in Atari games with a frameskip of 4. During both training and evaluation, EfficientZero chooses 150 simulations for each search in board games while 50 simulations of budget for Atari games. Other hyper-parameters are listed in Table 6.

### A.1.2  Training Details of Go

The detailed implementations of Atari games are discussed in EfficientZero [34]. However, it is nontrivial to adapt to board games. Here we give detailed instructions for training board games Go $9 \times 9$ in our implementations.

**Inputs** We follow the designs of AlphaZero, and we use the Tromp-Taylor rules, which is similar to previous work [31, 27]. The input states of the Go board are encoded into a $17 \times 9 \times 9$ array, which stacks the historical 8 frames and uses the last channels $C$ to identify the current player, 0 for black and 1 for white. Notably, the one historical frame consists of two planes $[X, Y]$, where the first plane $X$ represents the stones of the current player and the second one $[Y]$ represents the stones of the opponent. Besides, if there is a stone on board, then the state of the corresponding position in the frame will be

|  | **Go** $9 \times 9$ | **Atari** |
|---|---|---|
| Maximum number of tree size | 150 | 50 |
| Observation down-sampling | No | $96 \times 96$ |
| Total frames | 100M | 400k |
| Replay buffer size | 2M | 100k |
| Max frames per episode | 163 | 108k |
| Cost of training time | 24h | 8h |
| Komi of Go | 6.5 | - |
| Frame stack | 8 | 4 |
| Frame skip | 1 | 4 |
| Training steps | 100k | 100k |
| Mini batch | 256 | 256 |
| Learning rate | 0.05 | 0.2 |
| Weight decay ($c$) | 0.0001 | 0.0001 |
| Reward loss coefficient ($\lambda_1$) | 0 | 1 |
| Policy loss coefficient ($\lambda_2$) | 1 | 1 |
| Value loss coefficient ($\lambda_3$) | 1 | 0.25 |
| Consistency loss coefficient ($\lambda_4$) | 2 | 0.5 |
| Dirichlet $\alpha$ | 0.03 | 0.3 |
| $c_1$ in P-UCT | 1.25 | 1.25 |
| $c_2$ in P-UCT | 19652 | 19652 |
| $\epsilon$ | 0.1 | 0.1 |
| $r$ | 0.2 | 0.2 |

Table 6: Hyper-parameters of V-MCTS on Go $9 \times 9$ and Atari games

set to 1, otherwise to 0. For example, if the current player is black and suppose $b[i,j]$ is the current board state, $X[i,j] = \mathbf{1}_{b[i,j]=\text{black stone}}, Y[i,j] = \mathbf{1}_{b[i,j]=\text{white stone}}$. In summary, we concatenate together the historical planes to generate the input state $s = [X_{t-7}, Y_{t-7}, X_{t-6}, X_{t-6}, ..., X_t, Y_t, C]$, where $X_t, Y_t$ are the feature planes at time step $t$ and $C$ gives information of the current player.

**Training** As for the training phase, we train the model from scratch without any human expert data, which is the same as the setting of Atari games. Besides, limited to the GPU resources, we do not use the reanalyzing mechanism of MuZero [27] and EfficientZero [34], which targets at recalculation of the target values and policies from trajectories in the replay buffer with the current fresher model. Specifically, we use 6 GPUs for doing self-play to collect data, 1 GPU for training, and 1 GPU for evaluation.

**Exploration** To make a better exploration on Go, we reduce the $\alpha$ in the Dirichlet noise $\text{Dir}(\alpha)$ from 0.3 to 0.03, and we scale the exploration noise through the typical number of legal actions, which follows these works [31, 27]. In terms of sampling actions from MCTS visit distributions, we will mask the MCTS visit distributions with the legal actions and sample an action $a_t$, where

$$a_t := \begin{cases} a_t \sim \pi_t, & t < T \\ a_t = \arg\max \pi_t, & t \geq T \end{cases} \qquad (4)$$

$T$ is set to 16 in self-play and is set to 0 in evaluation, which is similar to these works [31, 27]. In this way, the agent does more explorations for the previous $T$ steps while taking the best action afterwards. But for Atari games, $\forall t$, we choose $a_t \sim \pi_t$ in self-play and $a_t = \arg\max \pi_t$ in evaluation, which is the same as these works [27, 34].

**Two-player MCTS** On board games, there are two players against each other, which is different from that of one-player games. Therefore, we should do some changes to the MCTS with the two-player game. For one thing, the value network always predicts the Q-value of the black player instead of the current player, which provides a more stable prediction. Furthermore, A significant change is that during backpropagation of MCTS, the value should be updated with the negative value from the child node. Because the child node is the opponent, the higher value of the opponent indicates a worse value of the current player. Besides, as for $Q$-values of the unvisited children on Go and Atari games,

we follow the implementation of EfficientZero [34] as follows:

$$\bar{Q}(s^{\text{root}}) = 0$$

$$\bar{Q}(s) = \frac{\bar{Q}(s^{\text{parent}}) + \sum_b \mathbf{1}_{N(s,b)>0}Q(s,b)}{1 + \sum_b \mathbf{1}_{N(s,b)>0}} \tag{5}$$

$$Q(s,a) := \begin{cases} Q(s,a) & N(s,a) > 0 \\ \bar{Q}(s) & N(s,a) = 0 \end{cases}$$

Notably, we allow the resignation for players when $\max_{a\in\mathcal{A}} Q(s^{root},a) < -0.9$ during self-play and evaluation, which means that the predicted winning probability is less than 5%. For convenience, when playing against GnuGo during evaluation, our agent will follow the skip action if GnuGo agent chooses the skip action. As for other hyperparameters on both Go and Atari games, we note that we choose the same values as those in EffcientZero. Specifically, the $c_1, c_2$ in our mentioned P-UCT formula (Eq. 1) are set to 1.25 and 19652, following these works [27, 34].

### A.1.3 Comparison of time cost on Go

To give the comparison of time cost among the methods considering the languages and hardwares. Here, we list the detailed settings of our models and the GnuGo engines in Table 7.

Table 7: Comparisons about the languages and hardware.

|        | C | Python | CPU | GPU | Time |
|--------|---|--------|-----|-----|------|
| MCTS   | ✓ | ✓      | ✓   | ✓   | 0.24 |
| V-MCTS | ✓ | ✓      | ✓   | ✓   | 0.12 |
| GnuGo  | ✓ |        | ✓   |     | 0.18 |

Table 8: Extra time consumed by virtual expansion on Go. Here $k$ is vanilla expansion times and $T$ is extra virtual expansion times.

|          | $T=30$ | $T=60$ | $T=90$ | $T=120$ |
|----------|--------|--------|--------|---------|
| $k=30$   | 0.7ms  | 1.5ms  | 2.2ms  | 3.0ms   |

To Give clear statistics about the extra time consumed by virtual expansion. Here, we record the time cost among different virtual expansion times in total after a fixed number of vanilla expansions. The results are listed in Table 8. We can find the extra time consumed by virtual expansion is little and linearly increased because there are only some atomic computations written in C++.

### A.2 Proof

Before the proof, let us recap some notations. In the MCTS procedure mentioned above, we suppose that there are total $|\mathcal{A}|$ actions to select and $N$ trials in total. Since the policy is defined as the visitation distributions of the root node, we only care about the $Q$-value and visitation changes of the root nodes.

Each action $a \in \mathcal{A}$ is associated with a value, which is a random variable bounded in the interval $[0, 1]$ with expectation $Q_a$. For convenience, we assume that different actions (arms) are ordered by their corresponding expected values, which means that $1 \geq Q_1 \geq Q_2 \geq \cdots Q_a \geq \cdots \geq Q_{|\mathcal{A}|} \geq 0$. At $k$ iteration step of the search loop, the agent will select an action at the root note and receive an independent sample of its value $R_a^k \in [0,1]$ from the neural networks, for simplification. And at $k$ step, the action $a \in \mathcal{A}$ of the root node is selected for $N_k(s,a) = T_a^k \leq k$ times of vanilla expansion. We use the notation $\bar{Q}_a^k = \frac{1}{T_a^k}\sum_{t=1}^{T_a^k} R_a^t$ to denote the empirical mean values and $U^k(s,a)$ to denote the ucb scores of the action $a$ given the root state $s$ at step $k$. (Here all $R_a^t$ are independent and bounded in $[0,1]$). For virtual expansion, we use $\hat{Q}_a^k$ to denote the empirical mean values at step $k$. It is obvious that $\hat{Q}_a^N = \hat{Q}_a^k$ when $k$ satisfies the VET-rule.

**Lemma A.1.** *Given $r \in (0,1)$, action set $\mathcal{A}$, $N > |\mathcal{A}|$. $\exists N_0, \forall N > N_0, k \geq rN$, we have $\forall a, T_a^k \geq 1$.*

*Proof.* The ucb score is defined by Eq. (1). Empirically, we will set $c_2$ given a budget $N$ and usually we have $c_2 > N \geq \sum_b N(s,b)$, so we note $c_1 + \log \frac{\sum_b N(s,b)+c_2+1}{c_2}$ as $c = c_1 + \log \frac{N+c_2+1}{c_2} \in (c_1, c_1 + \log 3)$ At step $k$, suppose there exist an action $a$, which has $T_a^k = 0$. Then at step $k$, the ucb score of $a$ should be

$$
\begin{aligned}
U^k(s,a) &= \bar{Q}(s) + P(s,a) \frac{\sqrt{\sum_b N(s,b)}}{1 + N(s,a)} c \\
&> \bar{Q}(s) + c_1 M_a \sqrt{k} \\
&> c_1 M_a \sqrt{k}, \text{ where } T_a^k = 0.
\end{aligned}
\tag{6}
$$

For action $b$, which has $T_b^k \geq 1$. At step $k$, we have

$$
\begin{aligned}
U^k(s,b) &= \bar{Q}_b^k + P(s,b) \frac{\sqrt{\sum_i N(s,i)}}{1 + N(s,a)} c \\
&< 1 + (c_1 + \log 3) M_b \frac{\sqrt{k}}{1 + T_b^k} \\
&< 1 + (c_1 + \log 3) M_b \frac{\sqrt{k}}{2}, \text{ where } T_b^k \geq 1.
\end{aligned}
\tag{7}
$$

Since $k \geq rN > rN - |\mathcal{A}|$ and $f(k) = c_1 M_a \sqrt{k} - (1 + (c_1 + \log 3) M_b \frac{\sqrt{k}}{2})$ is increasing for $k$. $\exists N_0$, we have $c_1 M_a \sqrt{rN_0} = 1 + (c_1 + \log 3) M_b \frac{\sqrt{rN_0}}{2}$. Let $N_0 = \max\{N_0, |\mathcal{A}|\}, \forall N > N_0, f(k) > 0$, $c_1 M_a \sqrt{k} > 1 + (c_1 + \log 3) M_b \frac{\sqrt{k}}{2}$. Then we have $U^k(s,a) > U^k(s,b)$. Therefore, at step $k$, for the action $b$ will be not selected. After extra $|\mathcal{A}|$ steps at most, all the action will be selected. Thus, we have $\forall a, T_a^k \geq 1$. $\qquad\square$

**Theorem A.2.** *(Value Consistency in Virtual Expansion): Given $r \in (0,1)$, confidence $\delta \in (0,1)$, finite action set $\mathcal{A}$. $\exists N_0, \forall N > N_0, k \geq rN$, let $\epsilon_k = \sqrt{\frac{1}{2k} \ln \frac{100k^2}{\delta}}$, we have (1) After $k$ times vanilla expansion, $Pr\{\bigcap_{a \in \mathcal{A}} |\bar{Q}_a^k - Q_a| < \epsilon_k\} > (1 - \frac{e\delta|\mathcal{A}|}{50k^2})$; (2) After $k$ times vanilla expansion and $N - k$ times virtual expansion, $Pr\{\bigcap_{a \in \mathcal{A}} \left|\hat{Q}_a^N - Q_a\right| < \epsilon_k\} > (1 - \frac{e\delta|\mathcal{A}|}{50r^2N^2})$, where $e$ is the Euler's number.*

*Proof.* Firstly, we have Hoeffding's inequality:

$$
\forall i = 1, 2, \cdots, n, a_i \leq X_i \leq b_i, S_n = X_1 + X_2 + \cdots + X_N
$$
$$
Pr\{|S_n - \mathbb{E}[S_n]| \geq t\} \leq 2 \exp \frac{-2t^2}{\sum_{i=1}^n (b_i - a_i)^2}
\tag{8}
$$

Observe that at step $k$, given confidence $\delta$, let $\epsilon_k = \sqrt{\frac{1}{2k} \ln \frac{100k^2}{\delta}}$, assumed that $R_a^t$ are independent and bounded in $[0,1]$. Then for action $a$, we have

$$
\begin{aligned}
Pr\{|\bar{Q}_a^k - \mathbb{E}[\bar{Q}_a^k]| \geq \epsilon_k\} &= Pr\{|\bar{Q}_a^k - Q_a| \geq \epsilon_k\} = Pr\{\left|\sum_{t=1}^{T_a^k} R_a^t - T_a^k Q_a\right| \geq \epsilon_k T_a^k\} \\
&\leq 2 \exp \left(\frac{-2(\epsilon_k T_a^k)^2}{T_a^k}\right) = 2 \exp \left(-2T_a^k \epsilon_k^2\right) = \delta_{k,a}
\end{aligned}
\tag{9}
$$

From Lemma A.1, we know that $1 \leq T_a^k \leq k$, we have $2 \exp \left(-2k\epsilon_k^2\right) \leq \delta_{k,a} \leq 2 \exp \left(-2\epsilon_k^2\right)$. After simplification, $\forall a \in \mathcal{A}$, we have

$$
\frac{\delta}{50k^2} \leq \delta_{k,a} \leq \frac{\delta}{50k^2} \exp \left(\frac{1}{k}\right)
\tag{10}
$$

And we know that $\sum_{a\in\mathcal{A}} T_a^k = k$, so we have

$$\begin{aligned}
Pr\{\bigcap_{a\in\mathcal{A}} |\bar{Q}_a^k - Q_a| \geq \epsilon_k\} &\leq \prod_{a\in\mathcal{A}} \delta_{k,a} \\
&= \prod_{a\in\mathcal{A}} 2\exp\left(-2T_a^k \epsilon_k^2\right) \\
&= 2\exp\left(-2k\epsilon_k^2\right) \\
&= \frac{\delta}{50k^2} = \delta_k
\end{aligned} \tag{11}$$

And from Eq. (9), we have $Pr\{|\bar{Q}_a^k - Q_a| < \epsilon_k\} \geq 1 - \delta_{k,a}$, then

$$\begin{aligned}
Pr\{\bigcap_{a\in\mathcal{A}} |\bar{Q}_a^k - Q_a| < \epsilon_k\} &\geq \prod_{a\in\mathcal{A}} (1 - \delta_{k,a}) \\
&\geq \prod_{a\in\mathcal{A}} (1 - \frac{\delta}{50k^2}\exp\left(\frac{1}{k}\right)) \\
&= (1 - \frac{\delta}{50k^2}\exp\left(\frac{1}{k}\right))^{|\mathcal{A}|}
\end{aligned} \tag{12}$$

Consider the function $f(x) = (1-x)^n - (1-nx), x = \frac{\delta}{50k^2}\exp\left(\frac{1}{k}\right) \in (0, \frac{e\delta}{50}], n = |\mathcal{A}| >= 2$, we have $f'(x) = -n(1-x)^{n-1} + n$. Since $\delta < 1, k \geq 1, x \leq \frac{e\delta}{50} < \frac{e}{50} < 1$. Then we have $f'(x) > 0$. Therefore, we have $f(x) > f(0) = 0$ and $(1-x)^n > (1-nx)$. So

$$\begin{aligned}
Pr\{\bigcap_{a\in\mathcal{A}} |\bar{Q}_a^k - Q_a| < \epsilon_k\} &= (1 - \frac{\delta}{50k^2}\exp\left(\frac{1}{k}\right))^{|\mathcal{A}|} \\
&> (1 - \frac{\delta|\mathcal{A}|}{50k^2}\exp\left(\frac{1}{k}\right)) \\
&> (1 - \frac{e\delta|\mathcal{A}|}{50k^2})
\end{aligned} \tag{13}$$

So we have

$$\begin{aligned}
\lim_{k\to\infty} \epsilon_k &= \lim_{k\to\infty} \sqrt{\frac{1}{2k}\ln\frac{100k^2}{\delta}} = 0, \\
\lim_{k\to\infty} \delta_k &= \lim_{k\to\infty} \frac{\delta}{50k^2} = 0, \\
\lim_{k\to\infty} (1 - \frac{e\delta|\mathcal{A}|}{50k^2}) &= 0,
\end{aligned} \tag{14}$$

Therefore, we know that

$$\begin{aligned}
Pr\{\bigcap_{a\in\mathcal{A}} |\bar{Q}_a^k - Q_a| < \epsilon_k\} &> (1 - \frac{e\delta|\mathcal{A}|}{50k^2}), \epsilon_k = \sqrt{\frac{1}{2k}\ln\frac{100k^2}{\delta}} \\
\lim_{k\to\infty} Pr\{\bigcap_{a\in\mathcal{A}} |\bar{Q}_a^k - Q_a| = 0\} &= 1
\end{aligned} \tag{15}$$

The probability can be converged to 1, and the convergence rate is $O(\frac{1}{k^2})$.

According to description of virtual expansion in Algo. 2, we know that after extra $N - k$ virtual expansion, the estimated $Q$-values keep the same as the $k$-step. This is because the visitation distributions of the previous $k$ steps are identical.

Therefore, for virtual expansion, the Eq. (15) is also satisfied.

Since $k \geq rN$, tor the next $N - k$ steps, the empirical mean $Q$-values $\hat{Q}_a^N$ are equal to $\bar{Q}_a^k$. So we have

$$Pr\{\bigcap_{a\in\mathcal{A}} |\hat{Q}_a^N - Q_a| < \epsilon_k\} > (1 - \frac{e\delta|\mathcal{A}|}{50k^2}) \quad > (1 - \frac{e\delta|\mathcal{A}|}{50r^2N^2}) \tag{16}$$

$\square$

**Theorem A.3.** *(Best Action Identification in Virtual Expansion): Given $r \in (0,1)$, confidence $\delta \in (0,1)$, finite action set $\mathcal{A}$. Suppose $\hat{Q}_a^N$ is the final empirical mean value of action after V-MCTS, $\bar{Q}_a^N$ is the final empirical mean value of after vanilla MCTS. $a = 1$ is the action of the highest expected value, $a = *$ is the action of the highest empirical mean value. $\exists N_0, \forall N > N_0, k \geq rN$, let $\epsilon_k = \sqrt{\frac{1}{2k} \ln \frac{100k^2}{\delta}}$, after V-MCTS, we have $Pr\{\left|\hat{Q}_*^N - \bar{Q}_1^N\right| < \epsilon_k + \epsilon_N\} > 1 - 2(\frac{\delta}{50k^2} \exp\left(\frac{1}{1.61\sqrt{k}}\right) + \frac{\delta}{50N^2} \exp\left(\frac{1}{N}\right))$.*

*Proof.* From Eq. (9) in Theorem A.2, we know that

$$Pr\{Q_1 - \bar{Q}_1^k < \epsilon_k\} \geq 1 - \delta_{k,1},$$
$$Pr\{\bar{Q}_1^k - Q_1 < \epsilon_k\} \geq 1 - \delta_{k,1} \tag{17}$$

, where $\epsilon_k = \sqrt{\frac{1}{2k} \ln \frac{100k^2}{\delta}}, \delta_{k,a} = 2 \exp\left(-2T_a^k \epsilon_k^2\right)$ Besides, we know that $\forall a \in \mathcal{A}, Q_a \leq Q_1$, so we have

$$Pr\{Q_1 - \bar{Q}_*^k < \epsilon_k\} \geq 1 - \delta_{k,*},$$
$$Pr\{\bar{Q}_*^k - Q_1 < \epsilon_k\} \geq 1 - \delta_{k,*} \tag{18}$$

For different step $k, N$, we have

$$Pr\{\bar{Q}_*^k - \bar{Q}_1^N < \epsilon_k + \epsilon_N\} \geq 1 - \delta_{k,*} - \delta_{N,1},$$
$$Pr\{\bar{Q}_1^k - \bar{Q}_*^k < 2\epsilon_k\} \geq 1 - \delta_{k,*} - \delta_{N,1} \tag{19}$$
$$\Rightarrow Pr\{\left|\bar{Q}_*^k - \bar{Q}_1^N\right| < \epsilon_k + \epsilon_N\} \geq 1 - 2(\delta_{k,*} + \delta_{N,1})$$

Before finding the bound of $\delta_{k,*}$, let us make an assumption first.

**Assumption A.4.** Suppose that $\forall a \in \mathcal{A}, M_a = P(s,a) \in (0,1)$ is the prior score obtained from the learned neural networks, we have $M_* \geq \frac{1}{|\mathcal{A}|} \sum_{a \in \mathcal{A}} M_a = \frac{1}{|\mathcal{A}|}$, where $* := \arg\max_a \bar{Q}_a^k$.

Here, this inequality is true when the learned neural networks can estimate the prior of the actions after training for some trials. In such a case, for the best empirical action $*$, the predicted prior score should be larger than the mean prior scores.

From the Lemma 2 in P-UCT [25], we know that at most $\frac{1.61\sqrt{n}}{M_*}$ distinct arms are pulled during the episode, where $M_*$ is the prior score $P(s,a)$ of the best action. $M_*$ is a constant during the search loop and we know that $M_* \geq \frac{1}{|\mathcal{A}|} \sum_{a \in \mathcal{A}} M_a = \frac{1}{|\mathcal{A}|}$. An action will be selected for more times with a higher empirical mean values. Therefore, for the empirical best action $*$, it has been selected more than $k/\frac{1.61\sqrt{k}}{M_*} = \frac{M_*}{1.61}\sqrt{k}$ times, which means $\frac{M_*}{1.61}\sqrt{k} \leq T_*^k \leq k$. So we have

$$\frac{\delta}{50k^2} \leq \delta_{k,*} \leq \frac{\delta}{50k^2} \exp\left(\frac{M_*}{1.61\sqrt{k}}\right) \tag{20}$$

From Eq. (20) and (10), we know that

$$Pr\{\left|\bar{Q}_*^k - \bar{Q}_1^N\right| < \epsilon_k + \epsilon_N\} \geq 1 - 2(\delta_{k,*} + \delta_{N,1})$$
$$\geq 1 - 2(\frac{\delta}{50k^2} \exp\left(\frac{M_*}{1.61\sqrt{k}}\right) + \frac{\delta}{50N^2} \exp\left(\frac{1}{N}\right)) \tag{21}$$

According to description of virtual expansion in Algo. 2, we know that after extra $N - k$ virtual expansion, the estimated $Q$-values keep the same as the $k$-step. Compared with the visitations of vanilla MCTS and V-MCTS, the only difference is the empirical mean $Q$-values. Observed that $\hat{Q}_a^N = \hat{Q}_a^k$ when $k$ satisfies the VET-rule.

Consequently, for the V-MCTS, we have

$$Pr\{\left|\hat{Q}_*^N - \bar{Q}_1^N\right| < \epsilon_k + \epsilon_N\} \geq 1 - 2(\frac{\delta}{50k^2} \exp\left(\frac{M_*}{1.61\sqrt{k}}\right) + \frac{\delta}{50N^2} \exp\left(\frac{1}{N}\right))$$
$$> 1 - 2(\frac{\delta}{50k^2} \exp\left(\frac{1}{1.61\sqrt{k}}\right) + \frac{\delta}{50N^2} \exp\left(\frac{1}{N}\right)) \tag{22}$$

$\square$

**Theorem A.5.** *(Error Bound of V-MCTS):* *Given $r \in (0,1)$, confidence $\delta \in (0,1)$, finite action set $\mathcal{A}$. Suppose the virtual expanded policy $\hat{\pi}_k$ is generated from Algorithm 3 (V-MCTS), $\exists N_0, \forall N > N_0, k \geq rN, \forall \epsilon \in [0,1]$, we have: if $\hat{\Delta}_s(k, k/2) < \epsilon$, $Pr\{||\pi_N(s) - \hat{\pi}_k(s)||_1 < 3\epsilon\} > 1 - \frac{e\delta|\mathcal{A}|}{50N^2}(1 + \frac{4}{r^2})$, where $e$ is the Euler's number.*

*Proof.* Suppose that $k$ satisfies the VET-rule, which means $k \geq rN, \epsilon \in [0,1], \hat{\Delta}_s(k, k/2) = \left|\left|\hat{\pi}_k(s) - \hat{\pi}_{k/2}(s)\right|\right|_1 < \epsilon$. Here, it is obvious that the given $\epsilon$ is in a range of $[0,1]$ because $\hat{\pi}_k(s)$ is a probability distribution.

In general, given the expected values $Q_a$ of each action $a$, assume there exists a ground truth policy $\pi(s)$, which does MCTS for N times given the expected values $Q_a$.

Then we have

$$\hat{\Delta}_s(N, k) = ||\hat{\pi}_N(s) - \hat{\pi}_k(s)||_1 \leq ||\hat{\pi}_N(s) - \pi(s)||_1 + ||\hat{\pi}_k(s) - \pi(s)||_1. \tag{23}$$

**Assumption A.6.** Suppose that given $\epsilon, r \in (0,1), \exists \sigma_\epsilon, N_0 > 0, \forall a \in \mathcal{A}, \forall N > N_0, k \geq rN$, when $\bigcap_{a \in \mathcal{A}} \left|\bar{Q}_a^k - Q_a\right| < \sigma_\epsilon$, we have $||\hat{\pi}_k(s) - \pi(s)||_1 < \epsilon$.

This assumption shows that when the L1 difference between all empirical mean values and the corresponding expected values, the difference of policy between $\pi(s)$ and $\hat{\pi}_k(s)$ can be bounded with the given distance $\epsilon$. This is obvious because virtual MCTS will do virtual expansion for the next $N - k$ times without changing the empirical mean values. Therefore, when $\sigma_\epsilon$ is small enough, during the next $N - k$ times expansion, the ucb scores of virtual expansion are similar to those of vanilla expansion with expected values. For example, when $\sigma_\epsilon \to 0, \bar{Q}_a^k \to Q_a$, the virtual expansion is totally the same as the vanilla expansion with expected values. Then $\exists N_0, \forall N > N_0$, $||\hat{\pi}_k(s) - \pi(s)||_1 = ||\hat{\pi}_N(s) - \pi(s)||_1 = 0 < \epsilon$.

From Eq. (15) in Theorem A.2, we have $Pr\{\bigcap_{a \in \mathcal{A}} \left|\bar{Q}_a^N - Q_a\right| < \epsilon_N\} > (1 - \frac{e\delta|\mathcal{A}|}{50N^2})$, where $\epsilon_N = \sqrt{\frac{1}{2N} \ln \frac{100N^2}{\delta}}$. Since $\exists N_1, \forall N > N_1, \sigma_\epsilon$ is a constant when $\epsilon$ is given, so $\sigma_\epsilon > \epsilon_N$, then with at least probability of $(1 - \frac{e\delta|\mathcal{A}|}{50N^2})$

$$||\hat{\pi}_N(s) - \pi(s)||_1 < \epsilon. \tag{24}$$

Since we know that $\left|\left|\hat{\pi}_k(s) - \hat{\pi}_{k/2}(s)\right|\right|_1 < \epsilon, ||\hat{\pi}_k(s) - \pi(s)||_1 - \left|\left|\hat{\pi}_{k/2}(s) - \pi(s)\right|\right|_1 \leq \left|\left|\hat{\pi}_k(s) - \hat{\pi}_{k/2}(s)\right|\right|_1 < \epsilon$.

From Eq. (15) in Theorem A.2, we have

$$Pr\{\bigcap_{a \in \mathcal{A}} \left|\bar{Q}_a^{k/2} - Q_a\right| < \epsilon_{k/2}\} > (1 - \frac{4e\delta|\mathcal{A}|}{50k^2}) \tag{25}$$

We know that $k \geq rN, \exists N_2, \forall N > N_2, \epsilon_k \leq \sqrt{\frac{1}{2rN} \ln \frac{100r^2N^2}{\delta}} < \sigma_\epsilon$, then with at least probability of $(1 - \frac{4e\delta|\mathcal{A}|}{50k^2})$, $\left|\left|\hat{\pi}_{k/2}(s) - \pi(s)\right|\right|_1 < \epsilon$ and

$$||\hat{\pi}_k(s) - \pi(s)||_1 \leq \left|\left|\hat{\pi}_{k/2}(s) - \pi(s)\right|\right|_1 + \left|\left|\hat{\pi}_k(s) - \hat{\pi}_{k/2}(s)\right|\right|_1 < 2\epsilon \tag{26}$$

Back to Eq. (23), with at least $(1 - \frac{e\delta|\mathcal{A}|}{50N^2}) \times (1 - \frac{4e\delta|\mathcal{A}|}{50k^2})$, we have

$$\begin{aligned}
\hat{\Delta}_s(N, k) &= ||\hat{\pi}_N(s) - \hat{\pi}_k(s)||_1 \\
&\leq ||\hat{\pi}_N(s) - \pi(s)||_1 + ||\hat{\pi}_k(s) - \pi(s)||_1 \\
&< \epsilon + 2\epsilon = 3\epsilon
\end{aligned} \tag{27}$$

For the $N$-th iteration of the search process, the final visitation distributions keep the same between the original expansion (Algorithm 1) and the virtual expansion (Algorithm 2). This is because at the

last iteration, searching the nodes after the root has no effects on the final distribution. Therefore, $\hat{\pi}_N(s) = \pi_N(s)$. So we have

$$
\begin{aligned}
||\pi_N(s) - \hat{\pi}_k(s)||_1 &= ||\hat{\pi}_N(s) - \hat{\pi}_k(s)||_1 \\
&\leq ||\hat{\pi}_N(s) - \pi(s)||_1 + ||\hat{\pi}_k(s) - \pi(s)||_1 \\
&< \epsilon + 2\epsilon = 3\epsilon
\end{aligned}
\tag{28}
$$

Therefore, let $N_0 = \max\{N_1, N_2\}, \forall N > N_0$,

$$
\begin{aligned}
Pr\{||\pi_N(s) - \hat{\pi}_k(s)||_1 < 3\epsilon\} &>= (1 - \frac{e\delta|\mathcal{A}|}{50N^2}) \times (1 - \frac{4e\delta|\mathcal{A}|}{50k^2}) \\
&> 1 - (\frac{e\delta|\mathcal{A}|}{50N^2} + \frac{4e\delta|\mathcal{A}|}{50k^2}) \\
&= 1 - \frac{e\delta|\mathcal{A}|}{50}(\frac{1}{N^2} + \frac{4}{k^2}) \\
&\geq 1 - \frac{e\delta|\mathcal{A}|}{50N^2}(1 + \frac{4}{r^2})
\end{aligned}
\tag{29}
$$

$\square$