# OpenReview forum: "Spending Thinking Time Wisely: Accelerating MCTS with Virtual Expansions"
_NeurIPS.cc/2022/Conference — NeurIPS 2022 Accept_

### Official Review · Reviewer_Gvr7 · 2022-07-10

**Rating:** 5
**Confidence:** 4
**Soundness:** 3 good
**Presentation:** 3 good
**Contribution:** 2 fair

**Summary:**

This paper proposes V-MCTS, containing two main improvements over classic MCTS. First, the authors propose virtual expansion by applying rollout without actual simulation (the simulation returns are replaced by the current Q value. Next, V-MCTS uses an adaptive termination condition to decide when to stop doing rollout. The proposed algorithm is evaluated on 9x9 Go and five Atari games.

**Questions:**

- How does virtual expansion compare with other action selection strategies (e.g., select by Q value, and some BAI strategies)?

- The empirical results are only tested with very few number of rollouts (<20 rollouts on Atari, and <150 on Go), and it remains unsure whether the proposed techniques are useful in more general cases.

**Limitations:**

The authors addressed the limitations and potential negative societal impact of their work.

**Strengths And Weaknesses:**

Improving the efficiency of MCTS is very important since they typically require a huge amount of computation resources. This paper proposes two techniques: virtual expansion and an early-termination condition.

Virtual expansion aims to mitigate the problem that exploratory behavior of the MCTS rollouts can "mask out" information about the action with the highest expected reward. For example, if the agent is given a budget of 100 rollouts and it has only figured out a promising path at the 95th rollout. The last 5 rollouts may not be sufficient to backpropagate this information up to the root node. In such cases, applying virtual expansion to backpropagate such information could be helpful. This is also justified by the ablation study in Sec. 5.4.

However, while virtual expansion could be useful in the above-mentioned scenario, I think it (i) could be implemented by an action selection policy for best-arm identification (BAI) and (ii) might only be useful in games with less chaotic rewards. (i): For example, it is possible that simply selecting the action that leads to the best cumulative reward seen in the rollouts can be as good as virtual expansion. (ii): For tasks where the reward function has high variance, it seems possible that virtual expansion will be fooled by some high rewards collected by chance. To allow better justification of the virtual expansion idea, it would be great if the authors could compare it with other action selection strategies (e.g., select by Q value, and some BAI strategies).

The theoretical analysis in the paper shows that V-MCTS will converge to the optimal policy as the number of rollouts increases, but it does not provide additional intuition on the comparison between vanilla MCTS and V-MCTS. Specifically, according to Thm 4.1, we still need k to be sufficiently large in order to guarantee e.g. BAI. Also, the theorems do not take into consideration the early-termination condition.

---

> ### Author Response · Authors · 2022-08-02
> **Response to Reviewer Gvr7**
>
> Thank you for your comments and advice! We hope the following address your concerns:
>
> As for the question "How does virtual expansion compare with other action selection strategies (e.g., select by Q value, and some BAI strategies)?":
>
> We would like to clarify that virtual expansion has two effects. (1) as you correctly mentioned, virtual expansions can mitigate the issue of exploratory behavior masking out the best action information. (2) However, virtual expansions do not aim to fully remove the exploratory behavior; instead, we aim to keep the exploratory component in the final policy. The oracle policy (without virtual expansions) is also highly exploratory in the early training phases, because of the Dirichlet exploration noise added and the inherent uncertainty of the model. Virtual expansion aims to keep that as well. We note that this is very important to RL, because without the exploratory part, it will quickly collapse due to over-exploitation issues.
>
> Moreover, we have a BAI ablation in our original paper in Section 5.4. The greedy expansion means that after k vanilla expansion, it will spend the left N − k simulations to visit the current best action greedily. But the performance is much poorer, which indicates that focusing on the best action leads to failure in hard-to-explore board games.
>
> As for the question "The empirical results are only tested with very few number of rollouts, and it remains unsure whether the proposed techniques are useful in more general cases": Thanks for the great question! To investigate whether our method still holds with larger amounts of MCTS expansions, we take a pretrained model and compare two strategies: (1) vanilla expansion with N=150/400/600/800 nodes in MCTS (2) virtual expanded policy with $N=800, r=0.2, \epsilon=0.1 $.
>
> |                | MCTS  ($N=150$) | MCTS  ($N=400$) | MCTS  ($N=600$) | MCTS ($N=800$) | V-MCTS ($N=800, r=0.2, \epsilon=0.1 $) |
> | -------------- | ------------- | ------------- | ------------- | ------------ | -------------------------------------- |
> | Average budget | 150           | 400           | 600           | 800          | 431.1                                  |
> | Winning rate   | 82.0%         | 84.5%         | 84.9%         | 85.9%        | 85.0%                                  |
>
> The result shows that (1) V-MCTS($N=800, r=0.2, \epsilon=0.1 $) is better than MCTS (N=600) in both the average budget and the winning rate, (2) V-MCTS can achieve comparable performance to the oracle MCTS(N=800) while keeping a much less average budget. Therefore, V-MCTS works with a larger amount of MCTS expansions.
>
> For the question of "might only be useful in games with less chaotic rewards", thanks for the insight! We agree that our method's usefulness will vary by the level of how chaotic the reward is. However, if we consider the game of Go, the reward function is already very chaotic. First, the reward function in the game of Go is only provided at the end of the game. The MCTS search can only be roughly guided by the value function, since the reward is all zero in the middle of the game. Second, the training in the game of Go is adversarial. This means that for any player, the environment is his opponent, who tries his/her best to screw up the first player. We empirically show that even in this sparse reward and adversarial setting, our V-MCTS is still useful. This provides strong evidence that our method is robust to chaotic rewards and environmental dynamics for many practical purposes.
>
> What's the intuition of the Theorem 4.1?
>
> Answer: As you have mentioned, the intuition is that V-MCTS converges to the optimal policy as the number of rollouts increases. Theorem 4.1 is a mathematically precise way of stating this intuition. Our theorem is a probabilistic statement, and one can choose the appropriate k given the desired confidence level. It is true that we need a sufficiently large k to absolutely guarantee BAI. However, in practice, we don't need it to be correct for every MCTS search. The theorem states that even if we can not guarantee the correct BAI, it still has a large probability of being correct.
>
> Finally, thanks for your suggestions! We have revised our paper and updated it on the website. And we highlight the changes and essential details that reviewers have mentioned with blue color.

---

### Official Review · Reviewer_7jdS · 2022-07-10

**Rating:** 7
**Confidence:** 5
**Soundness:** 4 excellent
**Presentation:** 3 good
**Contribution:** 3 good

**Summary:**

This paper proposed a novel method named Virtual MCTS (V-MCTS) to reduce the computation time of vanilla MCTS with the Virtual Expended Termination Rule (VET-Rule). This paper also gives theoretical bounds of the proposed method and evaluates the performance and computations on 9 × 9 Go board games and Atari games. Experiments show that this method can achieve comparable performances to the original search algorithm while requiring less than 50% search times on average. In general, this paper shows that V-MCTS can think faster on simple states and longer on hard states. For solid theoretical analysis and positive experimental results, I would recommend the acceptance of this paper.


**Questions:**

According to the proof given by the authors, V-MCTS’ Error Bound is still effective for programs that do not use virtual expansions during training. We should be able to apply V-MCTS to all pre-trained Top Go 19x19 programs, like KataGo or Leela Zero. How does V-MCTS work on these strong programs?

Some minor comments about presentations.
* In experiments, N = 150. I am just wondering what N_0 would be to satisfy Theorem 4.2, if N=150.
* It is unclear about ‘3 different runs’ in Tables 1~3. The authors need to clarify it.
* Please keep the consistency of the symbols, for example, $\epsilon$ and $eps$.
* Line 186: highlighted in line 5 -> highlighted in line 6
* In Algorithm 3, the symbols in line 10 are unclear and confusing to the reader. Explain why the ratio (the visitation of an action in a state divided by N) is the policy of this state. Where is that action? This conflicts with the situation in line 8.


**Limitations:**

Yes

**Strengths And Weaknesses:**

*Strengths*

* This paper gives the theoretical bounds of the proposed method with proof. Namely, for any given epsilon with sufficient large N, if the policy distance between k and k/2 is smaller than epsilon, then the distance between k and N is smaller than 3*epsilon. This theoretical result supports an early stop of MCTS search, which I believe has some impact. The theoretical analysis is sound, after I verify the proofs of Theorem 4.1 and 4.2 in Appendix (a little bit hard to read though).
* In the experiments with Go 9x9 and Atari, V-MCTS shows that this method can achieve comparable performances to the original search algorithm while requiring less than 50% search times on average.
* There is a comparison with DS-MCTS (a past work in AAAI), and V-MCTS still has better performance.

*Weaknesses*

Some minor presentation problems listed in Questions Section.

---

> ### Author Response · Authors · 2022-08-02
> **Response to Reviewer 7jdS**
>
> Thank you for your comments and corrections on the typos! We hope the following address your concerns:
>
> As for the question "apply V-MCTS to all pre-trained Top Go 19x19 programs", we have to mention that MuZero spends 16 TPUs for training and 1000 TPUs for selfplay in 19x19 Go games. Such computation resources are not available and affordable to our team. Therefore, we conduct the experiments among the Atari games and the harder Go 9x9 games to prove the effectiveness of V-MCTS.
>
> As for the question "In experiments, N = 150. What N_0 would be to satisfy Theorem 4.2, if N=150.", $N_0$ should satisfy the equation $c_1M_a \sqrt{rN_0} = 1 + (c_1 + \log 3)M_b \frac{\sqrt{rN_0}}{2}$ in Lemma A.1 in Appendix A.4 (line 688-689). It should be larger than $|A|$. In 9x9 Go games, $N_0$ should be larger than 82 (81 intersections as well as a `pass` action). In Figure 1 (b), V-MCTS (N=150) and (N=90) can work but V-MCTS(N=30) cannot because $N=30 < 82 \le N_0$. And we will add more explanations for this in the final version due to the limitation of the pages.
>
> As for the question "It is unclear about ‘3 different runs’ in Tables 1~3. The authors need to clarify it.", it means we do 3 separate training runs. We have revised our paper and updated it on the website.
>
> As for the suggestion "Please keep the consistency of the symbols, for example, $\epsilon$ and eps", we will add notations that the usage of $\epsilon$ is equal to eps as $\epsilon$ is hard to draw in figures.
>
> As for "Line 186: highlighted in line 5 -> highlighted in line 6", we have revised our paper and updated it on the website.
>
> As for the question "Explain why the ratio (the visitation of an action in a state divided by N) is the policy of this state. The symbols in line 10 conflict with the situation in line 8...Where is that action?": there is a typo in line 10 (Now is line 11) that causes some confusion. $\hat{\pi}_k(s) = \hat{N}_k(s, a) / N $ should be $\hat{\pi}_k(s, a) = \hat{N}_k(s, a) / N $. ${\pi}_k(s, a) \text{and} \hat{\pi}_k(s, a) $ are corresponding probabilities of action $a$, and we will sample an action from the returned policy distribution. As for why the visitation is divided by N, we have mentioned the definition of the policies in line 169. The policy distribution of MCTS is defined as the visitation of actions divided by the total visitations (line 9: $\pi_k(s, a) = N_k(s, a) / k $ for k-th iteration). But for the virtual expanded policy, it does $k$ vanilla expansion as well as $N - k$ virtual expansion, which means the total visitation is N rather than k.
>
> Finally, thanks for your detailed suggestions! We have revised our paper and updated it on the website. And we highlight the changes and important details that reviewers have mentioned with blue color.

---

### Official Review · Reviewer_wBBJ · 2022-07-12

**Rating:** 5
**Confidence:** 3
**Soundness:** 3 good
**Presentation:** 2 fair
**Contribution:** 2 fair

**Summary:**

This paper proposes a modification to the Monte-Carlo Tree Search paradigm---specifically, the UCT algorithm---that is more sample efficient. The method attempts to use its "thinking time" more wisely: as the authors describe it, their tree-building algorithm seeks to spend more time when evaluating "harder" states and less time on "simpler" states (i.e., states that are less positionally complex, and where the best action can be determined more easily). Identifying these situations is accomplished by noting when additional iterations of search would not appreciably change the visitation distribution of the actions at the root node (i.e., the stochastic policy at the root node). Specifically, on each iteration of tree-building, after the traditional loop of node selection-expansion-evaluation-value propagation is completed, some additional computation is performed -- what the authors term "virtual expansions". In this phase, $N-k$ "pulls" of the arms of the root node are performed, according to the selection strategy (for eg., UCB1), _but without descending down the tree further_ ($N$ = total search budget measured by iterations, $k$ = current iteration index). Instead, after each pull, the current average utiltity of that action $Q(s, a)$ is used as the reward, so that the only modification to the statistics accumulated in the nodes at level 1 in the search tree are in the visitation counts. The policy induced by the combination of "regular" tree-building and virtual expansions is tracked over time; when it begins to show signs of convergence (i.e., the norm between the policies from two sufficiently different iterations is sufficiently small), the search is terminated. The authors provide two theorems that characterize the nature of this convergence, as well as empirical evaluation in several domains---9x9 Go and five Atari games---to demonstrate the validity and benefit of their approach.

**Questions:**

(1) In all of the comparisons, V-MCTS was upper-bounded in how large a tree it could build. A complementary analysis where it was given a fixed _time_ budget would have been useful to see: to see if given a fixed amount of time, it could make better decisions than a vanilla MCTS using the same amount of time.

(2) The convergence criterion appears to rely on when the L1 norm of the policy at the root node converges; but I'm wondering if this is the best choice. There is an argument to be made for the L2 norm instead: bigger deviations should be amplified more, over an accumulation of smaller deviations, as this could change what action would be picked at the root. Or to ignore the specific distribution and focus on the _ordering_ of the nodes. It would be interesting to hear from the authors whether these alternatives were considered.

(3) Lines 58--59: "Afterward, UCT algorithms have generally replaced earlier heuristic methods for Monte-Carlo tree search (MCTS), which apply UCB1 to select action at each node of the tree" -- this framing not quite accurate. UCT is a specific instantiation of the MCTS family of algorithms, so saying it has "replaced" MCTS doesn't quite make sense.

(4) Lines 63--65: "There are three kinds of bottlenecks in vanilla MCTS in the aspect of speed: the selection stage of each iteration, the evaluation stage of each iteration, and the search loop." -- I found this confusing, since the search loop _includes_ selection and evaluation steps.

(5) Line 5 in Alg. 1, Line 9 in Alg. 2 -- I believe these should read UCB1(Q, P, N), rather than UCT(Q, P, N) (UCT is the overall algorithm, UCB1 (or a variant) is the bandit algorithm that is handling the selection step). Similarly, Line 4 in Alg. 3 -- should probably read "Selection with UCB1".

(6) In Theorem 4.1(a), what does the $\Cup$ (set intersection) mean? Did you mean $\Wedge$, i.e., an AND over the bounds on all possible actions?

(7) There are also typos and grammatical issues throughout the paper, that could be eliminated with a careful and thorough read through. Here's a small sampling just from the first page:
11--12: "while requiring less than 50% number of search times on average." --> "less than 50% search time"
19: "recent AI researches" --> "research"
26--27: "It is the first time for a computer program to beat a human professional Go player." --> "first time a computer program beat a human..."
29--30: "Later, MCTS-based RL algorithms are further extended to other board games and Atari games" --> Later, MCTS-based RL algorithms were further extended..."
32--33: "...they require massive computations to train and evaluate." ---> "...they require massive amounts of computation..."


**Limitations:**

Yes -- any concerns have been raised in other sections.

**Strengths And Weaknesses:**

Originality:
(+) The simplicity of the approach is a strength in my opinion. The idea is fairly straightforward, and the fact that it results in significant gains in a number of application domains is noteworthy.

Significance:
(+) MCTS is of broad interest in the ML and AI communities and papers that deal with enhancements to the baseline algorithm or propose novel applications are often published at NeurIPS. So the paper is likely to be of interest to many researchers.

Quality:
(+) The paper contains a nice balance of theory and experiment. The main theorems suggest that the authors' proposed approach should provide some benefits, and the empirical evaluation reinforces this.
(+) The gains in performance are particularly strong in 4/5 Atari domains, where V-MCTS (the authors' approach) outperforms vanilla MCTS while building much smaller search trees (typically, only 50% as big).
(+) The ablation experiments involving tuning the algorithm hyperparameters $r$ (the search budget ratio) and $\epsilon$ (which determines the tolerance criterion for convergence) were useful for evaluating their impact on performance.
(-) The results in 9x9 Go were however less compelling to me. The evidence here did not appear to support the authors' conclusions -- vanilla UCT was shown to be a little slower to act, but also a little stronger as a player. So claiming that V-MCTS was somehow preferable here, or a superior choice (265--266: "Therefore, such termination rule can keep strong performances with less budget.") is not borne out by the data.
(-) The qualitative analysis in Fig. 3 about V-MCTS spending more time in challenging positions and less time in less complex positions was welcome, but drawing such samples from a single game seems a little anecdotal. Aggregating information over many more such positions would have made for a stronger argument and paper.

Clarity:
Unfortunately, this was the weakest element of the paper, which relegates it to a borderline accept
in my eyes. The lack of clear writing and visualizations made it harder to evaluate some of the claims; it also raises questions about the reproducibility of the work. More specifics are provided in the next section, but I outline some broad issues here.
(-) No details are provided about the training procedure. EfficientZero is mentioned in Section 5.1, but it's not clear if that was the training procedure that was used. It's not clear if the code for reproducing the results will be made available.
(-) I found Fig. 1(a) difficult to understand, as it lumps together different algorithms that are being parameterized in different ways. I was also a little confused by why there was a _curve_ for GnuGo, which is an off-the-shelf Go playing engine -- presumably, the authors did not retrain this agent from scratch, so why are there different points along the $x$-axis for this player?
(-) The proofs for theorems 4.1 and 4.2 are presumably in the Appendix, so their proofs could not be verified; while I understand the authors are operating under space constraints, it would have been nice to at least see a proof sketch.

---

> ### Author Response · Authors · 2022-08-02
> **Response to Reviewer wBBJ**
>
> Thank you for your comments and advice! We hope the following address your concerns:
>
> For question (1) "Give a complementary analysis to see if given a fixed amount of time, it could make better decisions than a vanilla MCTS using the same amount of time.", we have made such a comparison in Figure 1(a). We will put more emphasis on this in the paper. In Figure 1(a), the red point of V-MCTS (eps=0.1) and the blue point of MCTS(N=90) consume the same amount of time (x-axis), but V-MCTS gives better performance (y-axis).
>
> For question (2) "There is an argument to be made for the L2 norm instead.": thanks for the great question! It is interesting to use the L2 norm of policy distributions. And here we make some ablations to see the difference. We take a pretrained model and compare two strategies (L1/L2 norm of distributions). And the results are as follows:
>
>
>
> |                | MCTS (N=150) | V-MCTS, **L1 norm** ($r=0.2, \epsilon=0.1$) | V-MCTS, **L2 norm** ($r=0.2, \epsilon=0.1$) | V-MCTS, **L2 norm** ($r=0.2, \epsilon=0.05$) |
> | -------------- | ------------ | ------------------------------------------- | ------------------------------------------- | -------------------------------------------- |
> | Average budget | 150          | **96.2**                                    | 97.1                                        | 119.3                                        |
> | Wining rate    | 82.0%        | **81.5%**                                   | 79.8%                                       | 81.0%                                        |
>
> We can find that (1) L2 norm can also work for V-MCTS; (2) L1 norm is better than L2 norm. And we attribute this to the formulation of ucb scores. Because the ucb scores have already taken into account the difference in the visitations (see the N(s, a) in Eq (1)). Therefore, amplifying the deviations may result in some bias.
>
> For suggestion (3), we agree that "this framing is not quite accurate". And we have revised our paper and updated it on the website.
>
> For suggestion (4), we agree with you and have committed the changes in the revised version: "The computation bottlenecks in vanilla MCTS come from the search loop, especially for the evaluation stage and the selection stage of each iteration.".
>
> For suggestion (5), we agree that we have confused UCT and UCB1 in writing sometimes. And we have revised our paper and updated it on the website.
>
> For question (6), the term `intersection` does mean "an AND over the bounds on all possible actions". We use intersection here to give a more simplified mathematical description.
>
> For suggestion (7), thank you for your detailed corrections. We have revised our paper and updated it on the website.
>
> Moreover, according to the questions in the Quality and Clarity parts, we make the following responses:
>
> As for the question "claiming that V-MCTS was somehow preferable here is not borne out by the data,": similar to the above answer to question (1), the comparison of V-MCTS(eps=0.1) and MCTS(N=90) can support the conclusion.
>
> As for the suggestion "Aggregating information over many more such positions would have made for a stronger argument and paper.", we agree with you and add more evaluations from more games (the agent plays as Black/White) in Appendix A.2.
>
> As for the question "No details are provided about the training procedure; It's not clear if the code for reproducing the results will be made available,": we have provided more detailed parameters in Appendix A.3 including models, hyper-parameters, and training details of Go. Moreover, we have mentioned in the checklist that the code will be available.
>
> As for the question "I was also a little confused by why there was a curve for GnuGo, which is an off-the-shelf Go playing engine", the GnuGo engine provides models of different levels (1-10). Each level is a trade-off between the run time and the strength of the agent. The y-axis is the winning rate against the model of level 10. We plot the green curve to show the performance-computation trade-off of the GnuGo engine and compare the trade-off with ours. Due to the limitation of pages, we will add more descriptions in the final version.
>
> As for the suggestion "it would have been nice to at least see a proof sketch.", we will add more descriptions in the final version.
>
> Finally, thanks for your detailed suggestions! We have revised our paper and updated it on the website. And we highlight the changes and essential details that reviewers have mentioned with blue color.

---

### Official Review · Reviewer_oRW8 · 2022-07-12

**Rating:** 3
**Confidence:** 4
**Soundness:** 2 fair
**Presentation:** 1 poor
**Contribution:** 2 fair

**Summary:**

The authors introduce Virtual MCTS which approximates its vanilla version with a smaller amount of computations. They also perform theoretical analysis as well as empirical performance analysis on 9x9 Go and the Atari game.


**Questions:**

Does Virtual MCTS attempt the time required for MCTS only when a model is trained? Or does it also attempt to reduce the time when playing a game against its opponent?

It apparently does not look Algorithms 1-2 are for the latter case.


**Strengths And Weaknesses:**

While the topic seems reasonable, I have a difficulty in understanding the problem they attempt to address (see the questions).  Typically game-playing programs have time management algorithms but the authors does not mention the approach.

For example,

https://dke.maastrichtuniversity.nl/m.winands/documents/time_management_for_monte_carlo_tree_search.pdf

https://www.remi-coulom.fr/Publications/TimeManagement.pdf

In addition, their comparison is against vanilla UCT. But another naive baseline is to stop UCT if the best move is much better than the others. It could be estimated by considering the number of visits and the reward received so far.

Misc.
Page 3: "Equation (1)"

Is this correct?

Page 3: "Thus, MCTS-based RL is roughly N times computationally more expansive than traditional RL algorithms"

expensive

Page 3: "It is consists of two components:"

It consists of

Page 4: "... illustrated in Algorithm 1, 2"

Algorithms 1 and 2

---

> ### Author Response · Authors · 2022-08-02
> **Response to Reviewer oRW8**
>
> Thank you for your comments and advice! We hope the following address your concerns:
>
>
> For the question "Typically game-playing programs have time management algorithms, but the authors do not mention the approach.": as we know, time management algorithms (TM) aim to do fine-grained time control conditioned by the total time cost of an episode. We think there are main distinctions between ours and time management algorithms:
>
> For one thing, the condition is different. TM algorithms are conditioned by a fixed time cost of an episode. Since the episodic length in board games is not constant, the thinking time of the current step is based on the time cost in the past. But V-MCTS targets normal games or tasks without any time constraints, which terminates the search based on the current situation of states.
>
> For the second, the target is different. TM algorithms aim to allocate time for each step in one episode (E.g., In a tournament). But V-MCTS aims at approximating the oracle policy distribution through early termination.
>
> Finally, the method is different. TM algorithms explicitly build dynamic strategies based on the real-time cost of the past steps. But V-MCTS or DS-MCTS would not take into account the time cost. Instead, we focus on policy distributions, not the past time cost or the left time budget.
> We will add more explanations for the distinctions.
>
> As for the suggestion "another naive baseline is to stop UCT if the best move is much better than the others.", we think it will cause severe exploration issues if only matching the best action without considering others or the entire distributions. This is because, in MCTS RL algorithms, the current policy not only needs to find which one is the best but also needs to maintain an exploration policy on the remaining potentially good actions. Actually, we did one ablation study in Sec. 5.4 where we greedily expand in MCTS, i.e., prioritizes the best action. But the greedy method fails due to weak exploration.
>
> As for the question "Does Virtual MCTS attempt the time required for MCTS only when a model is trained? Or does it also attempt to reduce the time when playing a game against its opponent?", our method reduces the time not only when a model is trained but also when the agent is playing a game. This is because V-MCTS can be applied if an evaluation model is given. Algorithms 1-2 display the detailed procedure of search iteration, and the search iteration is used when playing a game against its opponent. In conclusion, we propose the V-MCTS algorithm to reduce the search budget of MCTS adaptively while keeping comparable performances to the vanilla MCTS without early termination.
>
> As for the question "Misc. Page 3: 'Equation (1)'' Is this correct?", we confirm the correctness of the p-uct equation. This equation is the same as the paper: Mastering Atari, Go, Chess and Shogi by Planning with a Learned Model [1].
>
> As for other typos, we have revised our paper and updated it on the website.
>
> Finally, thanks for your detailed suggestions! We have revised our paper and updated it on the website. And we highlight the changes and essential details that reviewers have mentioned with blue color.
>
> [1] Schrittwieser, J., Antonoglou, I., Hubert, T., Simonyan, K., Sifre, L., Schmitt, S., ... & Silver, D. (2020). Mastering atari, go, chess and shogi by planning with a learned model. Nature, 588(7839), 604-609.

---

> ### Author Response · Authors · 2022-08-09
> **Discussion period ending soon**
>
> Dear reviewer oRW8,
>
> We kindly remind you that the final stage of discussion is ending soon, and so please kindly let us know if our response has addressed your concerns.
>
> Here is a summary of the revisions:
>
> - We further clarified the **main distinctions** between our work and the Time Management algorithms **from three aspects:** the conditions, the targets, and the method. They are not comparable and we will add a more detailed discussion on the related work in the final version.
>
> - We mentioned that we **made the ablation** of the different expansion methods, especially for the greedy one.
>
> - We **emphasized the targets of our work**. Namely, we aim to reduce the search budget of MCTS adaptively while keeping comparable performances to the vanilla MCTS without early termination.
>
> - We **revised** our paper and updated it on the website.
>
> Thanks again for your time and reviews, we will be happy to answer if there are additional issues or questions.

---

### Meta-Review · Area_Chair_T9rH · 2022-09-12

**Recommendation:** Accept
**Confidence:** Less certain

**Metareview:**

I found this to be an interesting paper.  As the reviewers indicated, it could be improved in terms of clarity, and I strongly encourage the authors to consider those comments carefully, as ultimately this could only make their paper more impactful.

In particular, the authors could consider how to be clearer about their claims, and how to provide stronger evidence for these.  For instance, a claim like "It can maintain comparable performances while reducing half of the time to search adaptively" is very general, and it is unclear that it is really true: for instance, is this true under _all_ conditions?

That said, I believe the paper is clear enough, and the method is simple enough, that it might be of interest to the community, and I think it would be good to accept it for presentation at the conference.  This agrees with most reviewers, three of whom voted to accept the paper.  I do agree with the one reviewer voting to reject that I'm somewhat unsure how this compares to other reasonable approaches, but I think this can be further discussed in follow-up papers as well.

**Award:**

No

---

### Decision · Program_Chairs · 2022-09-14

Accept